# Adaptive Symmetry Discovery for Dynamical System Identification

Behrooz Tahmasebi [1]   Melanie Weber [1]

## Abstract

Dynamical systems model trajectory data generated by underlying fixed dynamics, with applications ranging from biological systems to physics. Especially in scientific settings, dynamical systems are not generic, but exhibit symmetries imposed by physical laws, formalized as equivariance with respect to a group action. The identification problem concerns recovering the parameters of a system from observed trajectories. In this work, we study *adaptive symmetry discovery* for dynamical system identification and address how a system can be identified from a single trajectory when it is equivariant with respect to an unknown symmetry group. To this end, we first show that for known symmetries, the system can be identified from a significantly shorter single trajectory than in the generic setting, and we precisely characterize this improvement. We then consider the automatic symmetry discovery setting, proposing a method to learn the symmetry group directly from a single trajectory and incorporate it into the identification procedure, achieving the same optimal trajectory length as in the known-symmetry case. Our analysis relies on tools from group representation theory and the expander properties of Cayley graphs, and may be of independent interest for the study of symmetries in dynamical systems.

## 1. Introduction

The classical scientific approach has long relied on discovering fundamental laws of nature expressed through simple and interpretable equations. The recent abundance of data, together with rapid advances in machine learning and deep learning, has given rise to a complementary scientific paradigm: inferring governing laws and equations directly from data using principled methodologies rather than ad hoc modeling assumptions.

Among many modeling frameworks, dynamical systems constitute one of the most fundamental mathematical formalisms for describing flows, trajectory data, and non-independent evolutionary processes in the sciences, with applications ranging from biological systems to physical flows (Brunton et al., 2016; Yu & Wang, 2024; Wang & Yu, 2025). The problem of discovering governing equations from data, often formulated through the lens of dynamical systems (Brunton et al., 2016), has the potential to uncover previously unknown scientific principles and to significantly accelerate scientific discovery.

To this end, a central task is to learn the parameters of a dynamical system from data (observed trajectories), a problem commonly referred to as *dynamical system identification*. As a well-studied topic in system identification, a wide range of approaches and algorithms have been developed to address this problem across various settings (Van Overschee & De Moor, 2012; Isermann & Münchhof, 2011).

In the context of scientific discovery from data, however, dynamical systems are often not generic. Instead, they are structured and frequently exhibit (potentially unknown) symmetries imposed by physical laws, formalized as equivariance with respect to a (possibly unknown) group action. The field of geometric machine learning studies how and when such symmetries can be exploited to improve learning and generalization (Bronstein et al., 2021; Weber, 2025). In this work, we focus on the intersection of geometric machine learning and dynamical system identification.

Motivated by these considerations, we study the problem of *adaptive symmetry discovery* for dynamical system identification. Focusing on single-trajectory data and finite groups, the central question we address is how to identify a system when it is equivariant with respect to an unknown symmetry group, and how to automatically discover and exploit this symmetry to improve sample efficiency, namely by enabling identification from shorter trajectories.

First, we show that when the symmetry group is known, the system can be identified from significantly shorter trajectories than in the generic setting, and we precisely characterize

[1]Harvard John A. Paulson School of Engineering and Applied Sciences (SEAS), Harvard University, Cambridge, MA 02138, USA. Correspondence to: Behrooz Tahmasebi <behrooz_tahmasebi@seas.harvard.edu>.

*Proceedings of the 43rd International Conference on Machine Learning*, Seoul, South Korea. PMLR 306, 2026. Copyright 2026 by the author(s).

this improvement. Second, we turn to the automatic symmetry discovery setting and propose a method to learn the symmetry group directly from a single trajectory and to incorporate it into the identification procedure. Our approach achieves the same optimal trajectory length as in the known-symmetry case. Consequently, under mild conditions, symmetry discovery incurs negligible overhead, allowing one to fully leverage the benefits of equivariance.

It is worth noting that, while a number of recent studies are closely related to our work (as outlined in the related work section), most existing methods are heuristic or model-specific, and provable quantitative guarantees for symmetry discovery in dynamical systems are lacking. In contrast, our focus is on understanding the theoretical and foundational limits of this problem, which, to the best of our knowledge, have been largely unexplored in the literature.

Finally, we emphasize that the tools used in this paper, drawing from group representation theory and the expansion properties of Cayley graphs for finite groups, introduce techniques that are new to this context and may be of independent interest for the study of symmetries in dynamical systems and beyond.

In short, in this paper we make the following contributions:

- We study the problem of automatic symmetry discovery for dynamical system identification, with a focus on adapting to unknown finite group invariances.

- We show that symmetries can drastically reduce the trajectory length required to identify a system, and that this benefit can be achieved even when the symmetry is unknown by exploiting a new formulation that enables adaptation to the underlying symmetry.

- Our analysis leverages tools from group representation theory and the expander properties of Cayley graphs, which may be of independent interest for the broader study of symmetric and equivariant dynamical systems.

## 2. Related Work

Recently, the problem of learning dynamical systems from data has received particular attention, driven by applications across the sciences and physics-guided machine learning (Brunton et al., 2016; Yu & Wang, 2024; Wang & Yu, 2025; Sivaranjani et al., 2025). From a theoretical perspective, characterizing when and how a dynamical system can be identified from data is a classical and well-studied problem in system identification (Van Overschee & De Moor, 2012; Isermann & Münchhof, 2011). More recently, significant effort has focused on learning dynamical systems from noisy observations, a setting that goes beyond, and is substantially more challenging than, the noiseless identification regime. For example, Simchowitz et al. (2018) study

the identification of linear dynamical systems from a single noisy trajectory, with subsequent extensions to nonlinear systems under similar single-trajectory assumptions (Foster et al., 2020; Ziemann et al., 2022).

Beyond this, prior work has also investigated finite-sample identification of linear time-invariant systems (Sarkar & Rakhlin, 2019; Sarkar et al., 2021), regimes involving multiple trajectories (Tu et al., 2024), as well as system identification and control over a single trajectory (Fefferman et al., 2022; Carruth et al., 2022; 2024). In contrast, the focus of this paper is on the identification of dynamical systems *simultaneously* with symmetry discovery. As a first step toward this goal, we focus on the noiseless single-trajectory setting, which allows us to study the fundamental role of symmetry without additional statistical complications.

Although known symmetries are known to provide both empirical and theoretical benefits in learning and estimation, including improved generalization guarantees (Tahmasebi & Jegelka, 2023; 2024), in many applications the relevant symmetries exist but are not known *a priori*. In such settings, symmetries must be detected or discovered directly from data, as commonly encountered in the discovery of physical laws governed by differential equations. The research direction of *automatic symmetry discovery* seeks to identify underlying symmetries in a principled manner, moving beyond ad hoc or manually engineered approaches.

A wide range of methods for symmetry discovery have been proposed in the literature. Deep learning approaches for symmetry discovery have been explored in several works (Desai et al., 2022; Yang et al., 2023; Perin & Deny, 2025), including Lie algebra–based convolutional networks (Dehmamy et al., 2021); see also (Romero & Lohit, 2022; Ko et al., 2024; Hu et al., 2025b). Methods based on infinitesimal generators for discovering symmetries in nonlinear dynamical data have also been studied (Hu et al., 2025c); see (Shaw et al., 2025) for related approaches.

Beyond these, methods for discovering nonlinear group actions in latent spaces have been proposed (Yang et al., 2024a), including approaches that go beyond affine transformations and address manifold-structured data (Shaw et al., 2024; Bhat et al., 2025). Another line of work learns symmetries from layer gradients to relax hard invariance constraints (van der Ouderaa et al., 2023), while flow matching has recently been used to discover Lie group symmetries (Park et al., 2025). For symmetry discovery in differential equations and PDEs, see (Kreider et al., 2025; Hu et al., 2025a; Yang et al., 2025; 2024b). Other approaches include quadratic-form-based methods (Karjol et al., 2025) and learnable data augmentation strategies (Santos-Escriche & Jegelka, 2025). For symmetry discovery in finite groups using representation-theoretic tools, see (Huh, 2025). Jointly discovering and enforcing symmetries has also been studied

(Otto et al., 2025). Finally, we note that symmetry discovery is fundamentally different from (binary) hypothesis testing for the presence of symmetry in data (Soleymani et al., 2025b).

There has also been recent work on symmetry discovery, specifically in dynamical systems. Data-driven detection of Lie point symmetries for continuous dynamical systems is studied in Gabel et al. (2024), while the discovery of finite symmetry groups in dynamical systems is considered in Calvo-Barlés et al. (2025a;b). A related approach is proposed in Li et al. (2025), where the authors introduce latent mixtures of symmetries to model dynamical systems with multiple symmetric latent components.

Finally, we note that a variety of approaches have been proposed to introduce and exploit symmetries in learning, ranging from canonicalization (Kaba et al., 2023; Tahmasebi & Jegelka, 2025a;b; Shumaylov et al., 2025) and frame averaging (Puny et al., 2022; Atzmon et al., 2022; Lin et al., 2024) to data augmentation (Tahmasebi et al., 2025). In contrast, this paper introduces an approach based on generating sets of groups and their expansion properties, with close connections to recent work on approximate symmetry (Tahmasebi & Weber, 2025) and learning under invariances (Soleymani et al., 2025c;a).

# 3. Problem Statement

We first introduce the basic notation and formalize the class of dynamical systems studied in this paper. A detailed review of the required background is deferred to Appendix A.

**Dynamical systems.** A (discrete-time) dynamical system is specified by a function

$$f : \mathbb{C}^d \to \mathbb{C}^d,$$

which governs the state evolution according to

$$x_{t+1} = f(x_t), \qquad t = 0, 1, \ldots, \qquad (1)$$

where $x_t = (x_t^1, x_t^2, \ldots, x_t^d)^\top \in \mathbb{C}^d$ denotes the *state* of the system at time $t$. We refer to $f$ as the *dynamics* of the system and assume that $f$ belongs to a function class $\mathcal{F}$.

**Trajectories and learning objective.** Given an initial state $x_0 \in \mathbb{C}^d$, the dynamics $f$ generates a *trajectory*

$$(x_0, x_1, \ldots, x_T) \in \mathbb{C}^{d \times (T+1)},$$

where $T \in \mathbb{N}$ denotes the number of observed transitions. In the system identification problem, a learner observes such a trajectory and aims to recover the underlying dynamics $f \in \mathcal{F}$.

Clearly, if the function class $\mathcal{F}$ is too rich, recovering $f$ from a finite-length trajectory is impossible. This motivates restricting attention to structured and low-complexity families of dynamics.

## 3.1. Feature-lifted linear dynamical systems

In this work, we study a class of nonlinear dynamics that become linear after a finite-dimensional lifting of the state. Let

$$\Phi : \mathbb{C}^d \to \mathbb{C}^m$$

be an analytic feature map, and consider dynamics of the form

$$x_{t+1} = f(x_t) = W\Phi(x_t), \qquad W \in \mathbb{C}^{d \times m}. \qquad (2)$$

We refer to $\Phi$ as the *feature map* and to $\mathbb{C}^m$ as the *feature space*. For convenience, we write

$$\phi_t := \Phi(x_t) \in \mathbb{C}^m.$$

**Definition 3.1** (Feature-lifted linear dynamical systems). *Let $\Phi : \mathbb{C}^d \to \mathbb{C}^m$ be a fixed finite-dimensional feature map. We define*

$$\mathcal{F}_\Phi := \left\{ f : \mathbb{C}^d \to \mathbb{C}^d \mid f(x) = W\Phi(x) : \forall\, W \in \mathbb{C}^{d \times m} \right\}.$$

*Thus, every system in $\mathcal{F}_\Phi$ is nonlinear in the state $x$ in general, but linear in the lifted features $\Phi(x)$.*

This modeling assumption encompasses a rich family of nonlinear dynamical systems while retaining a linear parameterization in feature space. The choice of $\Phi$ determines the expressive power of the class, and later sections characterize how the representation-theoretic structure of the feature space controls the trajectory length needed for identifying the system.

**Example 3.2** (Polynomial features). A canonical choice is the polynomial feature map consisting of all monomials of total degree at most $k \in \mathbb{N}$:

$$\Phi_{\leq k}(x) = \left( (x^1)^{\alpha_1}(x^2)^{\alpha_2} \cdots (x^d)^{\alpha_d} \right)_{\alpha \in \mathcal{I}_k} \in \mathbb{C}^m,$$

where $\mathcal{I}_k = \left\{ \alpha \in \mathbb{Z}_{\geq 0}^d \mid \sum_{i=1}^d \alpha_i \leq k \right\}$. The feature dimension is $m = \binom{d+k}{d}$. We denote the corresponding class by $\mathcal{F}_{\leq k} := \mathcal{F}_{\Phi_{\leq k}}$. This recovers polynomial dynamical systems of total degree at most $k \in \mathbb{N}$.

**Example 3.3** (Fourier features). Another canonical choice is a finite band-limited Fourier feature map. Let $\Lambda_C \subset \mathbb{Z}^d$ be a finite set of frequencies, for instance $\Lambda_C = \{w \in \mathbb{Z}^d : \|w\|_2 \leq C\}$. Define

$$\Phi_C(x) = \left( e^{2\pi i \langle w, x \rangle} \right)_{w \in \Lambda_C} \in \mathbb{C}^{|\Lambda_C|}.$$

The corresponding class $\mathcal{F}_C := \mathcal{F}_{\Phi_C}$ consists of dynamics whose coordinates are finite trigonometric polynomials supported on $\Lambda_C$.

**Example 3.4** (Linear and affine systems)**.** Linear and affine dynamical systems are included as special cases of polynomial lifting. Indeed, a linear system

$$x_{t+1} = W_1 x_t, \qquad W_1 \in \mathbb{C}^{d \times d},$$

and an affine system

$$x_{t+1} = W_1 x_t + w_0, \qquad W_1 \in \mathbb{C}^{d \times d}, \ w_0 \in \mathbb{C}^d,$$

belong to $\mathcal{F}_{\leq 1}$, since $\Phi_{\leq 1}$ contains the constant feature and all coordinate functions.

**Example 3.5** (Nonlinear polynomial dynamics)**.** For $x = (x^1, x^2)^\top \in \mathbb{C}^2$, consider $f(x) = ((x^2)^2 - x^1 + 1, \ x^2 - x^1)^\top$. This system belongs to $\mathcal{F}_{\leq 2}$, since each coordinate is a polynomial of total degree at most 2.

## 3.2. Equivariant dynamics

We now introduce the notion of symmetry in dynamical systems. Intuitively, a dynamical system is symmetric if transformations applied to its input state induce predictable and consistent transformations of its output state. For example, rotating the input state results in a correspondingly rotated output. Such structure reflects underlying invariances often imposed by physical laws and is pervasive in scientific and engineering applications.

In this work, we formalize symmetry through the notion of *equivariance*. We begin by briefly reviewing the necessary group-theoretic background.

**Groups and representations.** A finite group $G$ is a finite set equipped with an associative binary operation, an identity element, and inverses for all elements. Common examples include finite rotation groups, reflection groups, and the permutation group on $n$ elements, consisting of all bijections $\sigma : [n] \to [n]$ under composition.

A linear representation of a group $G$ on $\mathbb{C}^n$ is a homomorphism $\rho : G \to \mathrm{GL}_n(\mathbb{C})$, assigning to each $g \in G$ an invertible matrix $\rho(g) \in \mathbb{C}^{n \times n}$ such that

$$\rho(gh) = \rho(g)\rho(h), \qquad \forall g, h \in G.$$

We refer to $\rho(g)$ as the action of $g$ on $\mathbb{C}^n$.

**Lifted group actions.** Suppose a finite group $G$ acts linearly on the state space $\mathbb{C}^d$ via a representation $\rho : G \to \mathrm{GL}_d(\mathbb{C})$. We assume that the feature map $\Phi : \mathbb{C}^d \to \mathbb{C}^m$ is compatible with this action, in the sense that there exists a representation $\rho_\Phi : G \to \mathrm{GL}_m(\mathbb{C})$ satisfying

$$\Phi(\rho(g)x) = \rho_\Phi(g)\Phi(x), \qquad \forall g \in G, \ x \in \mathbb{C}^d. \quad (3)$$

Thus, the group action on the state space induces a corresponding linear action on the feature space. This compatibility holds for the canonical feature maps considered in this paper, including polynomial features and finite Fourier features whenever the feature dictionary is closed under the action of $G$.

We are now ready to define equivariant dynamical systems.

**Definition 3.6** (Equivariant dynamical systems)**.** *Let $G$ be a finite group acting on $\mathbb{C}^d$ via a representation $\rho$. A dynamical system $f : \mathbb{C}^d \to \mathbb{C}^d$ is said to be $G$-equivariant if*

$$f(\rho(g)x) = \rho(g)f(x), \qquad \forall g \in G, \ x \in \mathbb{C}^d. \quad (4)$$

*For feature-lifted linear dynamical systems $f(x) = W\Phi(x) \in \mathcal{F}_\Phi$, $G$-equivariance is equivalently enforced by the intertwining condition*

$$\rho(g)W = W\rho_\Phi(g), \qquad \forall g \in G.$$

*We denote by $\mathcal{F}_\Phi^G$ the class of $G$-equivariant feature-lifted linear dynamical systems with feature map $\Phi$.*

This condition expresses the commutation of the parameter matrix $W$ with the group actions on the state and feature spaces, respectively.

*Remark* 3.7. Throughout the paper, we use the terms *symmetric* and *equivariant* interchangeably when referring to dynamical systems. The term equivariant is used when emphasizing the underlying group action.

## 3.3. Identifiability from a single trajectory

We now formalize the notion of identifiability.

**Definition 3.8** (Generic identifiability)**.** *Fix a feature map $\Phi : \mathbb{C}^d \to \mathbb{C}^m$, a finite group $G$, and consider the class $\mathcal{F}_\Phi^G$. The dynamics within this class are said to be* generically identifiable from trajectories of length $T \in \mathbb{N}$ *if, for almost all initial states $x_0 \in \mathbb{C}^d$ and almost all $G$-equivariant parameter matrices $W \in \mathbb{C}^{d \times m}$, the corresponding trajectory*

$$(x_0, x_1, \ldots, x_T) \in \mathbb{C}^{d \times (T+1)}$$

*uniquely determines $W$. The minimal such $T$ is denoted by $T_\Phi(G)$.*

Here, "generic" is understood in the standard sense: identifiability holds outside a set of measure zero in the space of initial states and $G$-equivariant parameters. This notion allows us to exclude pathological configurations while retaining full generality.

When identifiability holds for trajectories of length $T$, we informally refer to $T$ as the *sample complexity* of the identification problem, since each transition provides one observation of the dynamics.

*Remark* 3.9. The notation $T_\Phi(G)$ emphasizes that the trajectory length depends not only on the symmetry group $G$, but also on the feature map $\Phi$. Later, we characterize this

dependence through the representation-theoretic decomposition of the feature space and the generic excitation rank induced by $\Phi$.

**Symmetry discovery.** Consider an unknown $G$-equivariant dynamical system $f : \mathbb{C}^d \to \mathbb{C}^d$, where the underlying symmetry group $G$ is unknown. We assume, however, that $G$ belongs to a known class of admissible finite groups $\mathcal{G}$. While the specific group $G \in \mathcal{G}$ is not known a priori, prior knowledge of the class $\mathcal{G}$ provides structural information that can be leveraged for system identification.

We are primarily interested in settings where $\mathcal{G}$ consists of relatively large finite groups, as such symmetries can lead to substantial reductions in the trajectory length required for identifiability. At the same time, the unknown identity of $G$ introduces a nontrivial challenge, as the learner must simultaneously identify the dynamics and discover the underlying symmetry.

The goal of *adaptive symmetry discovery* for dynamical system identification is to recover the dynamics $f$ from short trajectories, using only the assumption that $f \in \mathcal{F}_\Phi$ is equivariant with respect to some unknown group $G \in \mathcal{G}$. In the ideal case, the best sample complexity one could hope to achieve is

$$T_\Phi(\mathcal{G}) := \max_{G \in \mathcal{G}} T_\Phi(G),$$

corresponding to the worst-case identifiability threshold over the class $\mathcal{G}$.

A further challenge arises from computational considerations. Finite groups of interest are often prohibitively large. For example, the permutation group on $n$ elements has cardinality $n! \approx \exp(n \log n)$, as do many of its subgroups. Consequently, algorithms whose runtime scales linearly with the group size are infeasible, and one must instead aim for procedures with runtime polylogarithmic in the group size. Accordingly, our aim is to address the following question:

> *Given a class of groups $\mathcal{G}$ and an unknown dynamical system $f : \mathbb{C}^d \to \mathbb{C}^d$ with $f \in \mathcal{F}_\Phi$ that is $G$-equivariant for some unknown $G \in \mathcal{G}$, can one identify the dynamics using trajectories of length $T_\Phi(\mathcal{G})$ while maintaining an efficient computational runtime?*

We answer this question affirmatively in the next section.

## 4. Main Results

In this section, we present the main theoretical results of the paper. We first characterize the trajectory length required for equivariant dynamical system identification when the symmetry group $G$ is known. We then build on this characterization to study adaptive symmetry discovery in the subsequent subsection.

### 4.1. Sample complexity of equivariant identification

We begin by studying $T_\Phi(G)$, the minimal trajectory length required to generically identify a $G$-equivariant system in $\mathcal{F}_\Phi^G$ when the finite group $G$ and the feature map $\Phi$ are known.

To state the result, we briefly recall a few standard notions from the representation theory of finite groups. A detailed background review is provided in Appendix A.

**Irreducible representations.** Let $G$ be a finite group, and let $\rho : G \to \mathbb{C}^{n \times n}$ be a linear representation. The representation $\rho$ is called *irreducible* if it has no nontrivial invariant subspace. The set of equivalence classes of irreducible representations of $G$ is finite and is denoted by $\widehat{G}$.

Any finite-dimensional representation decomposes, after a change of basis, into irreducible representations. In particular, we write

$$\mathbb{C}^d \cong \bigoplus_{\pi \in \widehat{G}} \mathbb{C}^{n_\pi} \otimes V_\pi, \qquad \mathbb{C}^m \cong \bigoplus_{\pi \in \widehat{G}} \mathbb{C}^{m_\pi} \otimes V_\pi,$$

where $V_\pi$ is the representation space of $\pi$, $d_\pi = \dim V_\pi$, and $n_\pi, m_\pi$ denote the multiplicities of $\pi$ in the state and feature representations, respectively.

Under this decomposition, a $G$-equivariant matrix $W : \mathbb{C}^m \to \mathbb{C}^d$ decomposes blockwise as

$$W = \bigoplus_{\pi \in \widehat{G}} C_\pi \otimes I_{V_\pi}, \qquad C_\pi \in \mathbb{C}^{n_\pi \times m_\pi}.$$

Thus, identifying $W$ amounts to identifying the multiplicity-space matrices $C_\pi$ for all irreps $\pi$ with $n_\pi > 0$.

**Generic excitation rank.** The representation decomposition alone does not determine the trajectory length: it also matters how the chosen feature map $\Phi$ excites the different isotypic components. For $x \in \mathbb{C}^d$, let $\Phi_\pi(x) \in \mathbb{C}^{m_\pi \times d_\pi}$ denote the $\pi$-isotypic component of $\Phi(x)$, reshaped according to $\mathbb{C}^{m_\pi} \otimes V_\pi$. For a trajectory $x_0, \ldots, x_T$, define

$$\oplus_{\pi,T} := [\Phi_\pi(x_0), \Phi_\pi(x_1), \ldots, \Phi_\pi(x_{T-1})] \in \mathbb{C}^{m_\pi \times T d_\pi}.$$

We define the *generic excitation rank* of the $\pi$-block at horizon $T$ by

$$h_{\pi,\Phi}(T) := \mathrm{rank}_{\mathrm{gen}}(\oplus_{\pi,T}),$$

where the generic rank is taken over the initial state and the equivariant parameter $W$ generating the trajectory. Equivalently, $h_{\pi,\Phi}(T)$ is the rank attained outside a measure-

zero set of trajectories. Since the entries of $\oplus_{\pi,T}$ are analytic functions of the generic parameters, this rank is well-defined.

We now state the main result characterizing the sample complexity of equivariant system identification.

**Theorem 4.1** (Equivariant system identification). *Fix a finite group $G$ and a feature map $\Phi : \mathbb{C}^d \to \mathbb{C}^m$. Consider the class $\mathcal{F}_\Phi^G$ of $G$-equivariant feature-lifted linear dynamical systems. Then $W$ is generically identifiable from a trajectory $x_0, \ldots, x_T$ if and only if*

$$h_{\pi,\Phi}(T) = m_\pi \qquad \text{for every } \pi \in \widehat{G} \text{ with } n_\pi > 0.$$

*Consequently, the sample complexity is*

$$T_\Phi(G) = \max_{\pi:n_\pi>0} \min\{T \in \mathbb{N} : h_{\pi,\Phi}(T) = m_\pi\}.$$

The theorem has a simple interpretation. In the $\pi$-isotypic block, the trajectory observations take the form

$$[X_{\pi,1}, \ldots, X_{\pi,T}] = C_\pi[\Phi_\pi(x_0), \ldots, \Phi_\pi(x_{T-1})].$$

Thus, the unknown matrix $C_\pi$ is identifiable exactly when the feature design matrix $\oplus_{\pi,T}$ has full row rank $m_\pi$. The quantity $h_{\pi,\Phi}(T)$ measures how many independent multiplicity directions are generically excited by a length-$T$ trajectory.

The ideal case occurs when each observation contributes the maximum possible number $d_\pi$ of independent directions in the $\pi$-block, namely when

$$h_{\pi,\Phi}(T) = \min\{m_\pi, Td_\pi\}.$$

In this case, Theorem 4.1 recovers the representation-theoretic threshold

$$T_{\text{ideal}}(G) = \max_{\pi:n_\pi>0} \left\lceil \frac{m_\pi}{d_\pi} \right\rceil.$$

This is the best possible threshold allowed by dimension counting, but it need not be achieved by every feature map.

More generally, define the one-step effective excitation dimension

$$r_{\pi,\Phi} := h_{\pi,\Phi}(1) = \text{rank}_{\text{gen}}\,\Phi_\pi(x).$$

Always $r_{\pi,\Phi} \leq d_\pi$. If the generic excitation rank grows linearly, i.e.,

$$h_{\pi,\Phi}(T) = \min\{m_\pi, Tr_{\pi,\Phi}\},$$

then the sample complexity simplifies to

$$T_\Phi(G) = \max_{\pi:n_\pi>0} \left\lceil \frac{m_\pi}{r_{\pi,\Phi}} \right\rceil.$$

Thus, the representation dimensions $d_\pi$ provide an ideal benchmark, while the effective dimensions $r_{\pi,\Phi}$ capture the actual excitation power of the chosen feature map.

This distinction is important for structured feature maps. For example, with polynomial features, repeated copies of the same irrep may arise as invariant multiples of the same covariant. In such cases, a single state may excite fewer than $d_\pi$ independent multiplicity directions, so $r_{\pi,\Phi} < d_\pi$ and more trajectory observations are needed. Theorem 4.1 therefore separates the representation-theoretic lower bound from the feature-dependent excitation profile that determines the actual identification threshold.

Finally, since $\sum_{\pi\in\widehat{G}} m_\pi d_\pi = m$, the ideal threshold can be much smaller than the ambient feature dimension $m$ when high-dimensional irreducible components are present. In the absence of symmetry, the trivial group has a single irrep with $d_\pi = 1$, and the threshold reduces to the usual feature-space requirement $T_\Phi(G) = m$ whenever the feature evaluations have generic full rank.

**Example 4.2** (Quadratic systems with permutation symmetry). Consider quadratic feature-lifted dynamics on $\mathbb{C}^d$ with permutation equivariance under the symmetric group $S_d$, where $d \geq 4$. Let $\Phi = \Phi_{\leq 2}$ be the feature map consisting of all monomials of total degree at most two. The feature space decomposes as

$$\mathcal{P}_{\leq 2} = V_0 \oplus V_1 \oplus V_2,$$

where $V_i$ denotes the space of homogeneous polynomials of degree $i$. The irreducible decompositions of these spaces are

$$V_0 = \pi_0,$$
$$V_1 = \pi_0 \oplus \pi_{\text{std}},$$
$$V_2 = 2\pi_0 \oplus 2\pi_{\text{std}} \oplus \pi_{(d-2,2)},$$

where $\pi_0$ is the trivial representation, $\pi_{\text{std}}$ is the standard representation with $d_{\pi_{\text{std}}} = d - 1$, and $\pi_{(d-2,2)}$ has dimension $d(d-3)/2$.

Thus, the feature multiplicities are

$$m_{\pi_0} = 4, \qquad m_{\pi_{\text{std}}} = 3, \qquad m_{\pi_{(d-2,2)}} = 1.$$

The state representation is the permutation representation $\mathbb{C}^d \cong \pi_0 \oplus \pi_{\text{std}}$, so only the trivial and standard irreps are active output blocks. For quadratic polynomial features, generic evaluations give $h_{\pi_0,\Phi}(T) = \min\{4, T\}$. For the standard block, one generically has $h_{\pi_{\text{std}},\Phi}(T) = \min\{3, 2T\}$: a single state excites two independent standard directions, rather than the ideal $d - 1$ directions, because one quadratic standard copy is an invariant multiple of the linear standard copy. Therefore

$$T_\Phi(S_d) = 4. \tag{5}$$

In contrast, the feature dimension is

$$m = 4 + 3(d-1) + \frac{d(d-3)}{2} = \Theta(d^2).$$

Thus, while $\Theta(d^2)$ trajectory length is required in the absence of symmetry, permutation equivariance reduces the sample complexity to a constant, independent of the state dimension $d$. This example also illustrates the role of the feature-dependent excitation profile: the ideal representation-theoretic bound would use $d_\pi$, whereas the actual threshold is governed by the generic ranks $h_{\pi,\Phi}(T)$.

Achieving the optimal feature-dependent bound $T_\Phi(G)$ without prior knowledge of the symmetry group $G$, which is the focus of the next subsection, is both challenging and highly appealing.

### 4.2. Adaptive symmetry discovery

We now turn to the problem of identifying equivariant dynamical systems when the underlying symmetry group is unknown. We recall the notion of generating sets, which allows for a compact representation of finite groups.

**Definition 4.3** (Generating set). *A subset $S \subseteq G$ of a finite group $G$ is called a* generating set *if every element $g \in G$ can be expressed as a finite product of elements of $S$, that is, $g = s_1 s_2 \cdots s_n$ for some $s_i \in S$. In this case, we write $G = \langle S \rangle$.*

Generating sets provide a succinct description of potentially large groups and play a central role in our symmetry discovery procedure. Unfortunately, finding a smallest generating set of a finite group is computationally difficult; this motivates the use of expander constructions to obtain reasonably small generating sets.

We are now ready to state the main result of the paper.

**Theorem 4.4** (Adaptive symmetry discovery). *Consider the problem of identifying an unknown dynamical system $f : \mathbb{R}^d \to \mathbb{R}^d$ that is equivariant with respect to an unknown finite group $G \in \mathcal{G}$, and $f \in \mathcal{F}_k^G$, for some $k \in \mathbb{N}$. Let $|G|_{\max} := \max_{G \in \mathcal{G}} |G|$, and fix a failure probability $\delta \in (0, 1)$.*

*There exists an algorithm (Algorithm 1) such that, given access to a single trajectory $(x_0, x_1, \ldots, x_T) \in \mathbb{R}^{d \times (T+1)}$ generated from a generic initial state by a generic $G$-equivariant system, the algorithm recovers the system and the symmetry group with probability at least $1 - \delta$ (over the algorithm's randomness), provided that*

$$T \geq T(\mathcal{G}) := \max_{G \in \mathcal{G}} T(G).$$

*Moreover, the algorithm returns a generating set $S$ such that $G = \langle S \rangle$. The runtime is polynomial in $m$, $|\mathcal{G}|$, $\log \frac{1}{\delta}$, and $\log |G|_{\max}$.*

---

**Algorithm 1** Adaptive symmetry discovery

---

1: **Input:** trajectory $(x_0, x_1, \ldots, x_T) \in \mathbb{R}^{d \times (T+1)}$, degree $k \in \mathbb{N}$ (or feature dimension $m = \binom{d+k}{d}$), failure probability $\delta \in (0, 1)$

2: **Output:** a matrix $W \in \mathbb{R}^{d \times m}$ and a generating set $S$ such that the dynamics is equivariant to $G = \langle S \rangle$

3: Compute features $\phi_t \leftarrow \Phi(x_t)$ for all $t = 0, \ldots, T-1$, and form

$$X \leftarrow [\phi_0, \phi_1, \ldots, \phi_{T-1}] \in \mathbb{R}^{m \times T},$$
$$Y \leftarrow [x_1, x_2, \ldots, x_T] \in \mathbb{R}^{d \times T}.$$

4: **for** each $G \in \mathcal{G}$ **do**

5:     Set

$$N \leftarrow 2.67\Big(\log |\mathcal{G}| + \log |G|_{\max} + \log \tfrac{1}{\delta} + \log 2\Big),$$
$$S \leftarrow \{g_1, \ldots, g_N\}, \quad g_i \overset{\text{iid}}{\sim} \text{Unif}(G).$$

6:     **Feasibility test:** check whether there exists $W \in \mathbb{R}^{d \times m}$ such that

$$WX = Y \quad \text{s.t.} \quad \rho(g)W = W\rho_\Phi(g), \ \forall g \in S.$$

7:     **if** feasible **then**

8:         Return $S$ and any feasible solution $W$, and terminate.

9:     **end if**

10: **end for**

---

We provide several remarks to clarify the implications of Theorem 4.4.

*Remark* 4.5 (Randomization). Algorithm 1 is probabilistic and depends on a failure probability $\delta \in (0, 1)$. It requires the ability to sample uniformly at random from the candidate groups $G \in \mathcal{G}$. The failure probability can be made arbitrarily small by increasing the number of sampled group elements, which scales logarithmically in $1/\delta$. Moreover, the algorithm can be made deterministic, with no probability of failure, if a small (i.e., logarithmic-size) generating set is given for each $G \in \mathcal{G}$. This case is simpler to handle, and we omit its proof, as it follows directly from the arguments presented in the paper.

*Remark* 4.6 (Optimal sample complexity). The algorithm achieves the *optimal sample complexity* for equivariant identification: as long as $T \geq T(\mathcal{G})$, a generic $G$-equivariant system can be identified. In addition, the algorithm successfully discovers the unknown symmetry by returning a generating set $S$ such that $G = \langle S \rangle$. Finding such an $S$ is essential for understanding how to project onto the space of $G$-equivariant dynamical systems, and allows one to fully test whether the dynamics is $G$-equivariant.

*Remark* 4.7 (Computational efficiency). Executing Algo-

rithm 1 involves checking feasibility of linearly constrained systems of linear equations and computing a solution whenever one exists. As shown in the proof, these tasks can be carried out in time polynomial in $(m, T, \log |G|_{\max})$. Importantly, the dependence on the group size is only logarithmic, which is essential for scalability.

Therefore, the main message of this paper can be summarized as follows:

> Equivariant dynamical systems can be identified from trajectories that are significantly shorter than those required in the generic, non-equivariant setting. Moreover, this reduction in sample complexity can be fully realized even when the symmetry group is unknown: the underlying symmetry can be discovered adaptively with only polylogarithmic computational overhead in the size of the finite group.

## 5. Proof Sketch

In this section, we provide a proof sketch of Theorem 4.4. We begin by examining the structure of Algorithm 1. The algorithm treats the observed trajectory as $T$ evaluations of the unknown dynamical system $f : \mathbb{R}^d \to \mathbb{R}^d$, which satisfies

$$x_{t+1} = W\Phi(x_t), \qquad t = 0, 1, \ldots, T-1. \quad (6)$$

Letting $\phi_t = \Phi(x_t)$ for all $t$, we define the data matrices

$$X = [\phi_0, \phi_1, \ldots, \phi_{T-1}] \in \mathbb{R}^{m \times T},$$
$$Y = [x_1, x_2, \ldots, x_T] \in \mathbb{R}^{d \times T}.$$

With this notation, identifying the dynamics reduces to finding a matrix $W \in \mathbb{R}^{d \times m}$ satisfying the linear system

$$WX = Y. \quad (7)$$

Any solution to Equation (7) corresponds to a dynamical system that is perfectly consistent with the observed trajectory. Consequently, the central questions are whether such a solution exists and, if so, whether it is unique.

Existence is immediate: by assumption, the trajectory is generated by a polynomially lifted linear dynamical system, and hence at least one solution $W$ must satisfy Equation (7). Uniqueness, however, is more subtle. In the generic (non-equivariant) setting, one might hope that $X$ is invertible, allowing the recovery $W = YX^{-1}$. This would require $T \geq m$, which is precisely the regime we seek to avoid. In the presence of symmetry, we aim to identify the system from trajectories of much shorter length.

To characterize all solutions of Equation (7), we invoke standard linear algebra. Let $X^\dagger \in \mathbb{R}^{T \times m}$ denote the Moore–Penrose pseudoinverse of $X$, which always exists and is

unique. Then the set of all solutions to Equation (7) is given by

$$W = YX^\dagger + Z(I_m - XX^\dagger), \qquad Z \in \mathbb{R}^{d \times m}, \quad (8)$$

where $Z$ is an arbitrary matrix. The term $YX^\dagger$ is the minimum-norm solution, while the second term spans the null space of the linear map $W \mapsto WX$. As $Z$ varies, Equation (8) parameterizes all matrices $W \in \mathbb{R}^{d \times m}$ consistent with the observed data.

The key insight underlying Algorithm 1 is that equivariance constraints significantly restrict the admissible set of matrices $W$ in Equation (8). By intersecting the affine solution space of Equation (7) with the linear subspace imposed by equivariance, the degrees of freedom collapse, yielding identifiability from short trajectories. The remainder of the proof formalizes this intuition and shows that sampling group elements suffices to recover the unknown symmetry with high probability.

For equivariant systems, it is generally the case that $XX^\dagger \neq I_m$, and therefore the solution to the linear system $WX = Y$ is not unique. In this regime, equivariance plays a crucial role. Since the true dynamics is equivariant with respect to a group $G$, the unknown matrix $W$ must satisfy

$$\rho(g)W = W\rho_\Phi(g), \qquad \forall g \in G. \quad (9)$$

Each such condition is linear in the entries of $W$, and collectively they impose additional constraints that can eliminate the spurious degrees of freedom in the solution set of $WX = Y$.

A direct implementation of these constraints, however, is computationally infeasible: enforcing equivariance for all $g \in G$ would require solving a system with $|G|$ linear constraints, which is prohibitive when $G$ is large. To overcome this issue, we leverage the notion of generating sets.

If $S \subseteq G$ is a generating set of $G$, written as $G = \langle S \rangle$, then equivariance with respect to $S$ already implies equivariance with respect to the entire group:

$$\rho(g)W = W\rho_\Phi(g), \quad \forall g \in S$$
$$\implies \rho(g)W = W\rho_\Phi(g), \quad \forall g \in G.$$

This implication follows directly from the definition of a generating set. Consequently, once a generating set $S$ is identified, imposing equivariance constraints only for elements of $S$ suffices to recover the unique $G$-equivariant solution $W$.

A key observation is that feasibility of the constrained system provides a test for equivariant to the unknown group $G$. Specifically, for any group $G \in \mathcal{G}$ with a generating set $S \subseteq G$, the system

$$WX = Y \quad \text{s.t.} \quad \rho(g)W = W\rho_\Phi(g), \quad \forall g \in S$$

is feasible if and only if the system is $G$-equivariant, provided that $T \geq T(\mathcal{G})$. This allows us to determine, using data alone, whether a sampled element belongs to the true symmetry group.

The core idea behind adaptive symmetry discovery is therefore as follows. We sample a collection of elements from any group $G \in \mathcal{G}$ to form a subset $S \subseteq G$ as a proxy for a generating set and then test each group for feasibility. When the number of samples $N$ is sufficiently large, we encounter elements of the subset $S \subseteq G$ form a generating set, with high probability.

Indeed, classical results on Cayley graph expanders (Alon & Roichman, 1994) imply that a random subset of a finite group $G$ of size $\mathcal{O}(\log |G| + \log \frac{1}{\delta})$ forms a generating set with probability at least $1 - \delta$. Combining this fact with the feasibility test described above and using a union bound shows that sampling

$$N = \mathcal{O}\Big( \log |\mathcal{G}| + \log |G|_{\max} + \log \tfrac{1}{\delta} \Big)$$

is sufficient to recover a generating set of the unknown symmetry group $G$, which in turn enables identification of the dynamics from short trajectories.

## 6. Conclusion and Future Work

In this paper, we study how to integrate adaptive symmetry discovery with the problem of dynamical system identification. Focusing on single-trajectory data, we address the question of how to identify a system that is symmetric with respect to an unknown finite group. Our main contribution is a method based on the theory of Cayley graph expanders and generating sets of finite groups, which enables adaptation to unknown symmetries with near-zero overhead. Moreover, the proposed approach achieves the same trajectory length (i.e., sample complexity) as in the setting where the symmetries are known, thereby yielding optimal adaptation.

An important direction for future work is to extend our results to infinite (Lie) groups, which would likely require theoretical tools beyond the expander-based framework developed for finite groups. Another promising direction is to study noisy dynamical systems and to identify system parameters while simultaneously adapting to symmetries in such settings. Establishing provable sample complexity guarantees in the presence of noise remains open, to the best of our knowledge. More broadly, it would be interesting to investigate whether similar adaptive symmetry techniques can be developed for learning linear time-invariant systems and for control systems with hidden states and noisy observations (Hazan et al., 2025). We leave these directions for future work.

## Acknowledgements

BT and MW were partially supported by NSF Award CBET-2112085 and DMS-2406905. MW acknowledges partial funding from an Alfred P. Sloan Fellowship in Mathematics and the AI2050 program at Schmidt Sciences (Grant G-25-69786).

## Impact Statement

This paper presents work whose goal is to advance the field of machine learning. There are many potential societal consequences of our work, none of which we feel must be specifically highlighted here.

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

# A. Preliminaries

In this section, we summarize the basic definitions and background material required to understand the results of the paper. Standard references for finite group theory and representation theory include (Serre et al., 1977; Fulton & Harris, 2013).

## A.1. Groups, representations, and equivariance

**Groups.** A (finite) group $G$ is a finite set equipped with a binary operation $\cdot : G \times G \to G$ satisfying the following properties:

- **Associativity:** $(g \cdot h) \cdot s = g \cdot (h \cdot s)$ for all $g, h, s \in G$.

- **Identity:** There exists an element $e \in G$ such that $e \cdot g = g \cdot e = g$ for all $g \in G$.

- **Inverses:** For every $g \in G$, there exists $g^{-1} \in G$ such that $g \cdot g^{-1} = g^{-1} \cdot g = e$.

For convenience, we drop the dot notation and write $gh$ instead of $g \cdot h$ for all $g, h \in G$.

The cardinality of a finite group $G$ is denoted by $|G|$.

**Examples of finite groups.** Here is a few examples of finite groups:

- The cyclic group of integers modulo $n$ under addition.

- The permutation group (also known as symmetric group) of group of all permutations of $n$ elements.

- Direct products of commutative groups, such as group of sign inversions $\{\pm 1\}^d$

- Dihedral group, consisting of symmetries of a regular polygon with $n$ sides in dimension two (reflections, rotations).

**Group actions and linear representations.** Let $V$ be a vector space over $\mathbb{C}$. A (left) *action* of a finite group $G$ on $V$ is a map

$$\theta : G \times V \to V$$

satisfying

$$\theta(e, v) = v, \qquad \theta(gh, v) = \theta(g, \theta(h, v)) \quad \text{for all } g, h \in G, \ v \in V.$$

When $\theta(g, \cdot)$ is linear for every $g \in G$, the action is equivalently described by a group homomorphism

$$\rho : G \to \mathrm{GL}(V),$$

where $\rho(g)v := \theta(g, v)$. In this case, $(V, \rho)$ is called a *linear representation* of $G$. Note that here each $\rho(g) \in \mathbb{C}^{m \times m}$ is an invertible matrix, with $m = \dim(V)$.

*Note:* While group actions can be defined on general sets or manifolds, throughout this paper we only consider linear actions on finite-dimensional complex vector spaces.

**Irreducible representations.** A representation $(V, \rho)$ of $G$ is said to be *irreducible* if it has no nontrivial $G$-invariant subspaces, i.e., the only subspaces $W \subseteq V$ satisfying $\rho(g)W \subseteq W$ for all $g \in G$ are $\{0\}$ and $V$ itself.

A fundamental result in finite group representation theory is *Maschke's theorem*, which states that every finite-dimensional representation of a finite group over $\mathbb{C}$ is completely reducible. That is, any representation $V$ admits a decomposition of the form

$$V \cong \bigoplus_{\pi \in \widehat{G}} \mathbb{C}^{n_\pi} \otimes V_\pi,$$

where:

- $\widehat{G}$ denotes the set of inequivalent irreducible representations of $G$,

- $V_\pi$ is a representative irreducible representation of dimension $d_\pi \in \mathbb{N}$,

- $n_\pi \in \mathbb{Z}_{\geq 0}$ is the multiplicity of $\pi$ in $V$.

The dimensions satisfy

$$\sum_{\pi \in \widehat{G}} n_\pi d_\pi = m.$$

Moreover, for finite groups, the number of irreducible representations is finite, thus $|\widehat{G}| < \infty$, and their dimensions obey the following identity

$$\sum_{\pi \in \widehat{G}} d_\pi^2 = |G|,$$

with each $d_\pi$ dividing $|G|$.

**Unitary representations and change of basis.** For finite groups, every finite-dimensional representation over $\mathbb{C}$ is equivalent to a unitary representation. In particular, after an appropriate change of basis, we may assume without loss of generality that $\rho(g)$ and $\rho_\Phi(g)$ are unitary for all $g \in G$. This change of basis does not affect equivariance properties or the structure of the state/feature space.

**Isotypic decomposition and block structure.** The decomposition above can be made explicit at the level of matrices. In particular, there exists a change of basis of $V$ under which the representation $\rho$ takes a block-diagonal form

$$\rho(g) = \bigoplus_{\pi \in \widehat{G}} \left( I_{n_\pi} \otimes \pi(g) \right), \qquad \forall g \in G,$$

where each $\pi(g) \in \mathbb{C}^{d_\pi \times d_\pi}$ is an irreducible representation and $I_{n_\pi}$ denotes the identity matrix of size $n_\pi$. We use the notation

$$\rho \cong \bigoplus_{\pi \in \widehat{G}} n_\pi \, \pi$$

to denote the decomposition of $\rho$ into irreducible representations, where $n_\pi$ denotes the multiplicity of $\pi$.

**Equivariant linear maps.** Let $(V, \rho_V)$ and $(U, \rho_U)$ be two representations of $G$. A linear map $\psi : V \to U$ is called *G-equivariant* if

$$\psi \circ \rho_V(g) = \rho_U(g) \circ \psi \quad \text{for all } g \in G.$$

The space of equivariant linear maps between $V$ and $U$ is denoted by $\mathrm{Hom}_G(V, U)$.

Using the decomposition into irreducible representations, the structure of $\mathrm{Hom}_G(V, U)$ can be characterized explicitly in terms of multiplicities. In particular, if

$$V \cong \bigoplus_{\pi \in \widehat{G}} \mathbb{C}^{n_\pi} \otimes V_\pi, \qquad U \cong \bigoplus_{\pi \in \widehat{G}} \mathbb{C}^{m_\pi} \otimes V_\pi,$$

then

$$\mathrm{Hom}_G(V, U) \cong \bigoplus_{\pi \in \widehat{G}} \mathbb{C}^{m_\pi \times n_\pi},$$

and in particular,

$$\dim(\mathrm{Hom}_G(V, U)) = \sum_{\pi \in \widehat{G}} m_\pi n_\pi.$$

This characterization will play a central role in our analysis of equivariant linear dynamical systems. In particular, it allows us to count the dimension of the matrices $W \in \mathbb{R}^{d \times m}$ that satisfy the equivariance condition in our proofs.

**Polynomial feature lifting.**   Fix a degree parameter $k \in \mathbb{N}$. Let

$$\Phi : \mathbb{R}^d \to \mathbb{R}^m$$

denote the polynomial feature map consisting of all monomials in $d$ variables of total degree at most $k$. Explicitly, for $x = (x^1, \dots, x^d)^\top \in \mathbb{R}^d$,

$$\Phi(x) \;=\; \left((x^1)^{\alpha_1}(x^2)^{\alpha_2} \cdots (x^d)^{\alpha_d}\right)_{\alpha \in \mathcal{I}_k}, \qquad \mathcal{I}_k = \left\{\alpha \in \mathbb{Z}_{\geq 0}^d \;\Big|\; \sum_{i=1}^d \alpha_i \leq k \right\}.$$

The resulting feature dimension is

$$m = \binom{d+k}{d}.$$

We refer to $\Phi(x)$ as the *feature representation* of the state $x$.

**Polynomially lifted linear dynamical systems.**   A *polynomially lifted linear dynamical system* of degree at most $k$ is a function

$$f : \mathbb{R}^d \to \mathbb{R}^d$$

of the form

$$f(x) = W\Phi(x), \qquad W \in \mathbb{R}^{d \times m}.$$

We denote by $\mathcal{F}_k$ the class of all such systems. Although $f$ is generally nonlinear in the state $x$, it is linear in the lifted feature space. Thus, each system in $\mathcal{F}_k$ is fully parameterized by the matrix $W$.

For convenience, along a trajectory $\{x_t\}_{t \geq 0}$, we write $\phi_t := \Phi(x_t) \in \mathbb{R}^m$.

**Lifted group actions on polynomial features.**   Suppose a finite group $G$ acts linearly on the state space $\mathbb{R}^d$ via a representation

$$\rho : G \to \mathrm{GL}_d(\mathbb{R}).$$

This action induces a natural linear action on the polynomial feature space associated with $\Phi$. In particular, there exists a unique representation

$$\rho_\Phi : G \to \mathrm{GL}_m(\mathbb{R})$$

such that

$$\Phi(\rho(g)x) = \rho_\Phi(g)\,\Phi(x), \qquad \forall g \in G, \; x \in \mathbb{R}^d. \tag{10}$$

The representation $\rho_\Phi$ corresponds to the action of $G$ on multivariate polynomials of degree at most $k$ induced by the change of variables $x \mapsto \rho(g)x$.

**Equivariant polynomially lifted systems.**   Let $G$ act on $\mathbb{R}^d$ via $\rho$. A dynamical system $f : \mathbb{R}^d \to \mathbb{R}^d$ is said to be *G-equivariant* if

$$f(\rho(g)x) = \rho(g)f(x), \qquad \forall g \in G, \; x \in \mathbb{R}^d.$$

For polynomially lifted linear systems $f(x) = W\Phi(x) \in \mathcal{F}_k$, this condition is equivalent to the matrix constraint

$$\rho(g)W = W\rho_\Phi(g), \qquad \forall g \in G, \tag{11}$$

which states that $W$ is an intertwining operator between the representations $\rho_\Phi$ and $\rho$.

We denote by $\mathcal{F}_k^G \subseteq \mathcal{F}_k$ the class of all $G$-equivariant polynomially lifted linear dynamical systems of degree at most $k$.

**Equivariance as linear constraints.**   For fixed representations $\rho$ and $\rho_\Phi$ of a finite group $G$, the equivariance condition

$$\rho(g)W = W\rho_\Phi(g), \qquad \forall g \in G,$$

is a system of linear constraints in the entries of $W$. Consequently, the space of $G$-equivariant matrices forms a linear subspace of $\mathbb{R}^{d \times m}$ whose dimension can be characterized using representation-theoretic multiplicities given above. This observation is essential for the study of the time complexity of optimization problems considered later in the paper.

**Genericity and polynomials.** Throughout the paper, we use the term *generic* to refer to properties that hold outside a set of Lebesgue measure zero (equivalently, outside a proper algebraic variety). In particular, a vector $z \in \mathbb{R}^m$ is said to be generic if it does not satisfy any nontrivial polynomial equations beyond those imposed by the problem structure.

Polynomial feature maps inherit strong genericity properties. Let $\Phi : \mathbb{R}^d \to \mathbb{R}^m$ denote the polynomial lifting map consisting of all monomials up to degree $k$. Then, for almost every $x \in \mathbb{R}^d$, the feature vector $\Phi(x)$ is generic in $\mathbb{R}^m$ in the sense that it avoids all proper algebraic subvarieties of the feature space.

More generally, let $\mathcal{P}(\Phi(x))$ denote any finite collection of polynomial constraints in the entries of $\Phi(x)$. If these constraints do not vanish identically as polynomials in $x$, then the set of points $x \in \mathbb{R}^d$ for which $\mathcal{P}(\Phi(x)) = 0$ has Lebesgue measure zero.

This notion of genericity is preserved under polynomial transformations. In particular, if $x$ is generic and $f : \mathbb{R}^d \to \mathbb{R}^d$ is a polynomial map, then $\Phi(f(x))$ is also generic, except possibly on a measure-zero set. Consequently, along trajectories generated by polynomially lifted linear dynamical systems, the lifted feature vectors $\Phi(x_t)$ remain generic almost surely.

These genericity properties will be used repeatedly to justify rank conditions and to rule out degenerate algebraic configurations in our proofs. That is where the assumption of polynomial feature spaces becomes useful.

## A.2. Cayley graphs and expanders

Let $G$ be a finite group. For a multiset $S = \{s_1, \ldots, s_N\}$ of elements of $G$, the (left) *Cayley graph* $\mathrm{Cay}(G, S)$ is the $|G|$-vertex (multi)graph with vertex set $G$ and directed edges $g \to s_i g$ for each $g \in G$ and $i \in [N]$. When $S$ is symmetric (i.e., $S = S^{-1}$ as a multiset), the graph can be viewed as an undirected $N$-regular multigraph. Its normalized adjacency (averaging) operator is

$$\mathcal{A}_S f(g) := \frac{1}{N} \sum_{i=1}^{N} f(s_i g), \qquad f : G \to \mathbb{C}.$$

The graph is an *expander* if $\mathcal{A}_S$ has a spectral gap: the trivial eigenvalue 1 (corresponding to constant functions) is separated from the rest of the spectrum. We will use a standard representation-theoretic characterization of this gap.

**Representation-theoretic block diagonalization.** Consider the left-regular representation of $G$ on $\ell_2(G)$, and the operator $\mathcal{A}_S$ above. Over $\mathbb{C}$, $\ell_2(G)$ decomposes into irreducible representations (irreps) as

$$\ell_2(G) \cong \bigoplus_{\pi \in \widehat{G}} d_\pi \, V_\pi,$$

where $V_\pi$ is the irrep space of dimension $d_\pi$ and the multiplicity of $\pi$ in the regular representation equals $d_\pi$. Under this decomposition, $\mathcal{A}_S$ becomes block diagonal with blocks indexed by $\pi \in \widehat{G}$, and the block corresponding to $\pi$ is (unitarily equivalent to)

$$A_\pi(S) := \frac{1}{N} \sum_{i=1}^{N} \pi(s_i) \in \mathbb{C}^{d_\pi \times d_\pi}.$$

In particular, the second-largest (in magnitude) eigenvalue of $\mathcal{A}_S$ equals

$$\lambda(\mathcal{A}_S) = \max_{\pi \neq \mathbf{1}} \|A_\pi(S)\|_{\mathrm{op}},$$

where $\mathbf{1}$ denotes the trivial irrep.

**From spectral gap to generation.** If $\langle S \rangle$ denotes the subgroup generated by $S$, then $\mathrm{Cay}(G, S)$ is connected if and only if $\langle S \rangle = G$. For symmetric $S$, connectivity is equivalent to having no additional eigenvalue 1 besides the trivial one. The following lemma provides a useful sufficient condition.

**Lemma A.1** (Spectral certificate for generation). *Let $S$ be a (multi)set in $G$. If*

$$\max_{\pi \neq \mathbf{1}} \|A_\pi(S)\|_{\mathrm{op}} < 1,$$

*then $\langle S \rangle = G$ (equivalently, $\mathrm{Cay}(G, S)$ is connected).*

*Proof.* If $\langle S \rangle \neq G$, let $H = \langle S \rangle$ be a proper subgroup. Consider the permutation representation of $G$ acting on left cosets $G/H$. The associated (normalized) averaging operator over $S$ fixes every function that is constant on each $H$-coset, hence has eigenvalue 1 on a subspace of dimension $|G/H| > 1$. Removing the global constants yields a nontrivial invariant subspace on which the eigenvalue is still 1. Decomposing this representation into irreps, one finds a nontrivial irrep $\pi \neq \mathbf{1}$ for which $A_\pi(S)$ has eigenvalue 1, and thus $\|A_\pi(S)\|_{\mathrm{op}} = 1$, contradicting the assumption. $\qquad\square$

**Random generators via matrix Bernstein.** We now quantify the probability that $N$ uniform samples generate $G$. Let $g_1, \ldots, g_N \overset{\text{iid}}{\sim} \mathrm{Unif}(G)$, and set $S = \{g_1, \ldots, g_N\}$. Fix a nontrivial irrep $\pi \neq \mathbf{1}$, and consider

$$A_\pi(S) \; = \; \frac{1}{N} \sum_{i=1}^{N} \pi(g_i).$$

We may assume $\pi$ is unitary. Since $g_i$ is uniform and $\pi$ is nontrivial, Schur orthogonality implies

$$\mathbb{E}\,\pi(g_i) = \frac{1}{|G|} \sum_{g \in G} \pi(g) = 0.$$

Thus $X_i := \pi(g_i)$ are independent, mean-zero, and satisfy $\|X_i\|_{\mathrm{op}} = 1$. Moreover $X_i X_i^* = I_{d_\pi}$, hence

$$\left\| \sum_{i=1}^{N} \mathbb{E}\left[ X_i X_i^* \right] \right\|_{\mathrm{op}} = \| N I_{d_\pi} \|_{\mathrm{op}} = N.$$

Applying a standard matrix Bernstein inequality (e.g., for rectangular/complex matrices) yields:

$$\Pr\left( \|A_\pi(S)\|_{\mathrm{op}} \geq 1 \right) \; \leq \; 2 d_\pi \exp\left( -\frac{3N}{8} \right). \tag{12}$$

In particular, this gives an exponentially small tail in $N$. See (Tahmasebi & Weber, 2025) for more details on the proof.

**Union bound over all irreps.** By Lemma A.1, it suffices to ensure $\|A_\pi(S)\|_{\mathrm{op}} < 1$ for all nontrivial $\pi$. Fix any target failure probability $\delta \in (0, 1)$. Using Equation (12) and a union bound,

$$\mathbb{P}\left( \exists \pi \neq \mathbf{1} : \|A_\pi(S)\|_{\mathrm{op}} \geq 1 \right) \; \leq \; \sum_{\pi \neq \mathbf{1}} 2 d_\pi \exp\left( -\frac{3N}{8} \right).$$

Using the crude bound $\sum_{\pi \in \widehat{G}} d_\pi \leq |G|$ (since $d_\pi \leq d_\pi^2$ and $\sum_\pi d_\pi^2 = |G|$), we obtain

$$\mathbb{P}\left( \exists \pi \neq \mathbf{1} : \|A_\pi(S)\|_{\mathrm{op}} \geq 1 \right) \; \leq \; 2|G| \exp\left( -\frac{3N}{8} \right). \tag{13}$$

**Quantitative guarantee for random generation.** Combining Lemma A.1 with Equation (13) yields the following.

**Proposition A.2** (Random samples generate $G$ with high probability). *Let $g_1, \ldots, g_N \overset{\text{iid}}{\sim} \mathrm{Unif}(G)$ and $S = \{g_1, \ldots, g_N\}$. Then*

$$\mathbb{P}\left( \langle S \rangle \neq G \right) \; \leq \; \mathbb{P}\left( \exists \pi \neq \mathbf{1} : \|A_\pi(S)\|_{\mathrm{op}} \geq 1 \right) \; \leq \; 2|G| \exp\left( -\frac{3N}{8} \right).$$

*Consequently, to ensure $\mathbb{P}(\langle S \rangle \neq G) \leq \delta$, it suffices to take*

$$N \; \geq \; 2.67 \Big( \log |G| + \log(1/\delta) + \log 2 \Big)$$

.

*Remark* A.3. The explicit failure bound Equation (13) is especially convenient when applying a union bound over many candidate groups (e.g., $G \in \mathcal{G}$). If $N$ samples are drawn independently for each $G$, then choosing

$$N \; \geq \; 2.67 \Big( \log |\mathcal{G}| + \log |G|_{\max} + \log(1/\delta) + \log 2 \Big)$$

ensures success simultaneously for all groups with probability at least $1 - \delta$, where $|G|_{\max} := \max\limits_{G \in \mathcal{G}} |G|$.

# B. Proofs of Main Results

We provide the proof of the main theorems in the paper here in this section.

## B.1. Proof of Theorem 4.1

*Proof.* We prove the claim by characterizing the degrees of freedom of $G$-equivariant polynomially lifted linear systems and determining the minimum trajectory length required to identify them uniquely from a single trajectory.

**Step 1: Structure of equivariant parameters.** By $G$-equivariance, the parameter matrix $W \in \mathbb{R}^{d \times m}$ satisfies

$$\rho(g)W = W\rho_\Phi(g), \qquad \forall g \in G.$$

As reviewed in Appendix A, this condition implies that $W$ is an intertwining operator between the representations $\rho_\Phi$ and $\rho$. Under a suitable change of basis, we may assume that the lifted representation decomposes as

$$\rho_\Phi \cong \bigoplus_{\pi \in \widehat{G}} \left( I_{n_\pi} \otimes \pi \right).$$

By Schur's lemma, the space of equivariant linear maps is block-diagonal across irreducible components, and each block corresponding to $\pi$ has the form

$$W_\pi = C_\pi \otimes I_{d_\pi}, \qquad C_\pi \in \mathbb{R}^{d \times n_\pi}.$$

Consequently, the unknown parameters associated with the irrep $\pi$ are encoded in the matrix $C_\pi$, which has $d \times n_\pi$ free parameters.

**Step 2: Observations as linear constraints.** Consider a trajectory $\{x_t\}_{t=0}^T$ generated by

$$x_{t+1} = W\Phi(x_t).$$

Let $\phi_t := \Phi(x_t)$. Each observation yields the linear constraint

$$x_{t+1} = W\phi_t.$$

Projecting onto the $\pi$-isotypic component of the feature space, this equation reduces to

$$x_{t+1}^{(\pi)} = \left( C_\pi \otimes I_{d_\pi} \right) \phi_t^{(\pi)}.$$

Equivalently, each time step provides $d_\pi$ linearly independent equations in the entries of $C_\pi$, provided that $\phi_t^{(\pi)}$ is generic.

**Step 3: Genericity and rank growth.** By the genericity assumptions on the initial state $x_0$ and the matrix $W$, and by the genericity properties of polynomial feature maps established in Appendix A, the lifted features $\{\phi_t\}_{t \geq 0}$ are generic almost surely. In particular, for each $\pi$, the sequence $\{\phi_t^{(\pi)}\}_{t \geq 0}$ spans a generic subspace of the $\pi$-isotypic component.

As a result, after $T$ time steps, the total number of independent linear constraints on $C_\pi$ is at most $T \times d_\pi$. To uniquely identify $C_\pi \in \mathbb{R}^{d \times n_\pi}$, we must therefore have

$$T \times d_\pi \geq n_\pi.$$

Thus, identification of the $\pi$-component requires

$$T \geq \left\lceil \frac{n_\pi}{d_\pi} \right\rceil, \quad \forall \pi \in \widehat{G}.$$

**Step 4: Worst-case irrep governs sample complexity.** Since the parameters corresponding to different irreducible representations are independent, the overall trajectory length must be sufficient to identify the most demanding component. Therefore, the sample complexity of $G$-equivariant generic system identification is

$$T(G) = \max_{\pi \in \widehat{G}} \left\lceil \frac{n_\pi}{d_\pi} \right\rceil.$$

This completes the proof. $\qquad \square$

*Remark* B.1. When $G$ is trivial, $\rho_\Phi$ is the trivial representation with multiplicity $m$, and the bound reduces to the classical requirement $T \geq m$, recovering the generic (non-equivariant) sample complexity.

## B.2. Proof of Theorem 4.4

*Proof.* We prove the theorem by establishing correctness, sample complexity, and runtime guarantees for Algorithm 1. The proof proceeds in several steps.

**Step 1: Solving linear systems under equivariance constraints.** Fix a candidate group $G \in \mathcal{G}$ and a sampled set $S = \{g_1, \ldots, g_N\} \subseteq G$. Given the observed trajectory $(x_0, \ldots, x_T)$, define

$$X = [\phi_0, \phi_1, \ldots, \phi_{T-1}] \in \mathbb{R}^{m \times T}, \qquad Y = [x_1, x_2, \ldots, x_T] \in \mathbb{R}^{d \times T},$$

where $\phi_t = \Phi(x_t)$. The feasibility test in Algorithm 1 determines whether there exists a matrix $W \in \mathbb{R}^{d \times m}$ satisfying the linear system

$$WX = Y, \tag{14}$$

subject to the equivariance constraints

$$\rho(g_n)W = W\rho_\Phi(g_n), \qquad n = 1, \ldots, N, \tag{15}$$

where $\rho(g_n)$ and $\rho_\Phi(g_n)$ are known representation matrices.

**Step 2: Vectorized formulation.** To obtain a unified linear-algebraic characterization, we work with the vectorization operator $\text{vec}(\cdot)$. Let

$$w := \text{vec}(W) \in \mathbb{R}^{dm}.$$

Using the standard identity

$$\text{vec}(WX) = (X^\top \otimes I_d)\,\text{vec}(W),$$

the data constraint in Equation (14) becomes

$$(X^\top \otimes I_d)w = \text{vec}(Y). \tag{16}$$

Similarly, using

$$\text{vec}(AWB) = (B^\top \otimes A)\,\text{vec}(W),$$

the equivariance constraints in Equation (15) are equivalent to

$$\left(I_m \otimes \rho(g_n) - \rho_\Phi(g_n)^\top \otimes I_d\right)w = 0, \qquad n = 1, \ldots, N. \tag{17}$$

Let $C \in \mathbb{R}^{(Ndm) \times (dm)}$ denote the matrix obtained by stacking the left-hand sides of Equation (17). Then the original constrained system is equivalent to finding $w \in \mathbb{R}^{dm}$ satisfying

$$(X^\top \otimes I_d)\,w = \text{vec}(Y), \qquad Cw = 0. \tag{18}$$

**Step 3: Characterization of the equivariant solution space.** The homogeneous system $Cw = 0$ defines a linear subspace

$$\mathcal{N}(C) := \{w \in \mathbb{R}^{dm} : Cw = 0\},$$

which contains exactly the vectorizations of matrices $W$ satisfying all equivariance constraints. Let $Z \in \mathbb{R}^{dm \times r}$ be a matrix whose columns form a basis of $\mathcal{N}(C)$, where $r := \dim \mathcal{N}(C)$. Every equivariant solution admits the parameterization

$$w = Z\theta, \qquad \theta \in \mathbb{R}^r. \tag{19}$$

**Step 4: Reduction to an unconstrained linear system.** Substituting Equation (19) into Equation (16) yields

$$(X^\top \otimes I_d)Z\theta = \text{vec}(Y). \tag{20}$$

Thus, a feasible equivariant solution exists if and only if $\text{vec}(Y)$ lies in the column space of $(X^\top \otimes I_d)Z$, i.e.,

$$\text{rank}\big((X^\top \otimes I_d)Z\big) = \text{rank}\Big(\big[(X^\top \otimes I_d)Z \;\; \text{vec}(Y)\big]\Big).$$

**Step 5: Correctness for the true group.** Let $G^\star$ denote the true symmetry group of the dynamics. By construction, the true system matrix $W^\star$ satisfies Equation (15) for all $g \in G^\star$. By the expander-based sampling result in Appendix A, choosing

$$N = \mathcal{O}\big(\log |\mathcal{G}| + \log |G|_{\max} + \log(1/\delta)\big)$$

ensures that, with probability at least $1 - \delta$, the sampled set $S \subseteq G$ generates $G$, for all $G \in \mathcal{G}$, including $G = G^\star$. In this event, the null space $\mathcal{N}(C)$ coincides with the space of all $G^\star$-equivariant matrices.

Since the initial state and the system matrix are generic, and $T \geq T(G^\star)$, Theorem 4.1 implies that the reduced system in Equation (20) admits a unique solution $\theta^\star$, corresponding to $W^\star$. Hence, the feasibility test succeeds for $G^\star$ and the algorithm returns a generating set $S$ with $G^\star = \langle S \rangle$.

**Step 6: Failure for incorrect groups.** Consider any candidate group $G \neq G^\star$. In this case, $\mathcal{N}(C)$ consists of matrices that are $G$-equivariant but not $G^\star$-equivariant. By genericity of the trajectory and of polynomial features, the probability that $\mathrm{vec}(Y)$ lies in the column space of $(X^\top \otimes I_d)Z$ is zero. Thus, the reduced system in Equation (20) is infeasible almost surely, and the algorithm correctly rejects all incorrect groups.

**Step 7: Sample complexity.** By Theorem 4.1, identification of a $G$-equivariant system requires $T \geq T(G)$. Therefore, requiring

$$T \geq T(\mathcal{G}) \coloneqq \max_{G \in \mathcal{G}} T(G)$$

ensures correctness uniformly over $\mathcal{G}$.

**Step 8: Computational complexity.** Let $p \coloneqq dm$. Constructing each constraint operator

$$C_n = I_m \otimes \rho(g_n) - \rho_\Phi(g_n)^\top \otimes I_d$$

requires $\mathcal{O}(p^2)$ operations. Forming all $N$ constraints costs $\mathcal{O}(Nd^2m^2)$ time and memory.

Stacking these constraints and computing a basis for $\mathcal{N}(C)$ via standard linear algebra requires

$$\mathcal{O}(Np^3) = \mathcal{O}(Nd^3m^3)$$

time in the worst case. Forming the reduced system costs $\mathcal{O}(dTpr)$, and solving it requires

$$\mathcal{O}(dTr^2 + r^3),$$

which in the worst case yields $\mathcal{O}(d^3m^2T + d^3m^3)$.

**Overall complexity.** Combining the above steps, the total runtime per group is bounded by

$$\mathcal{O}\big(Nd^3m^3 + d^3m^2T\big),$$

with memory complexity $\mathcal{O}(Nd^2m^2)$. Since the algorithm loops over $|\mathcal{G}|$ groups and

$$N = \mathcal{O}\left(\log |\mathcal{G}| + \log |G|_{\max} + \log \tfrac{1}{\delta}\right),$$

the overall runtime is polynomial in $m$, $|\mathcal{G}|$, and $\log |G|_{\max}$. This completes the proof. $\qquad\square$

## C. Experiments

In this section, we present a simple proof-of-concept experiment that illustrates the theoretical results developed in this paper. Though our focus is on foundational aspects, this experiment serves as a complementary empirical validation of our analysis.

*Table 1.* Dimension of the space of equivariant maps $\{A \in \mathbb{R}^{d \times d} : \rho(g)A = A\rho(g), \ \forall g \in G\}$ for groups actions on $\mathbb{R}^d$ (with $d = 10$).

| Group $G$ | Generators | $\dim$(equivariant maps) | Description |
|---|---|---|---|
| Trivial (no symmetry) | $\{e\}$ | $d^2 = 100$ | All linear maps |
| Single transposition $C_2 = \langle(1\,2)\rangle$ | one swap | $d^2 - 2d + 2 = 82$ | Weak symmetry constraint |
| Cyclic shifts $C_d$ | one $d$-cycle | $d = 10$ | Circulant matrices |
| Dihedral group $D_d$ | shift + reversal | $\frac{d}{2} + 1 = 6$ | Symmetric circulant |
| Block permutations $S_5 \times S_5$ | within-block swaps | $6$ | Two symmetric blocks |
| Full symmetric group $S_d$ | adjacent swaps | $2$ | $\alpha I + \beta \mathbf{1}\mathbf{1}^\top$ |

**Dimension of equivariant maps for permutation symmetries.** Throughout this experiment, we consider linear dynamics on $\mathbb{R}^d$ of the form

$$x_{t+1} = Ax_t, \qquad t = 0, 1, \ldots, T-1, \tag{21}$$

where $A \in \mathbb{R}^{d \times d}$ is unknown. Let $G$ be a finite group that acts on $\mathbb{R}^d$ by permuting coordinates, represented by permutation matrices $\rho(g) \in \mathbb{R}^{d \times d}$ for $g \in G$. We say that the dynamics is $G$-equivariant if

$$\rho(g)\, A \;=\; A\, \rho(g), \qquad \forall g \in G. \tag{22}$$

The set of all matrices satisfying Equation (22) is a linear subspace of $\mathbb{R}^{d \times d}$, sometimes called the *centralizer* (or *commutant*) of the action of $G$:

$$\mathcal{C}(G) \;:=\; \{A \in \mathbb{R}^{d \times d} : \rho(g)A = A\rho(g) \text{ for all } g \in G\}.$$

For permutation actions, the dimension $\dim(\mathcal{C}(G))$ has a simple interpretation. Indeed, Equation (22) is equivalent to the entrywise invariance condition

$$A_{ij} = A_{g(i)\,g(j)} \qquad \forall g \in G,$$

meaning that entries of $A$ must be constant on the orbits of $G$ acting on ordered index pairs $(i,j) \in \{1, \ldots, d\} \times \{1, \ldots, d\}$. Consequently,

$$\dim(\mathcal{C}(G)) \;=\; \#\Big(\text{orbits of } G \text{ acting on } \{1, \ldots, d\} \times \{1, \ldots, d\}\Big). \tag{23}$$

Table 1 reports $\dim(\mathcal{C}(G))$ for several standard choices of $G$ when $d = 10$. For example, when $G = S_d$ is the full symmetric group acting by all coordinate permutations, there are only two orbits (diagonal pairs $(i,i)$ and off-diagonal pairs $(i,j)$ with $i \neq j$), hence $\dim(\mathcal{C}(S_d)) = 2$ and every $S_d$-equivariant matrix has the form $\alpha I + \beta \mathbf{1}\mathbf{1}^\top$ for some $\alpha, \beta \in \mathbb{C}$.

**Dimension of the feasible solution set vs. trajectory length.** Given a single trajectory $(x_0, x_1, \ldots, x_T)$ generated by Equation (21), define the data matrices

$$X := [x_0, x_1, \ldots, x_{T-1}] \in \mathbb{R}^{d \times T}, \qquad Y := [x_1, x_2, \ldots, x_T] \in \mathbb{R}^{d \times T}.$$

The data constraint is $Y = AX$. Fix an *assumed* symmetry group $H$ acting by permutations, and restrict $A$ to lie in the linear subspace $\mathcal{C}(H)$. We study the corresponding feasible set of solutions

$$\mathcal{S}_T(H) \;:=\; \{A \in \mathcal{C}(H) : Y = AX\}. \tag{24}$$

Whenever non-empty, $\mathcal{S}_T(H)$ is an affine subspace of $\mathbb{R}^{d \times d}$. We report its affine dimension, denoted $\dim(\mathcal{S}_T(H))$. In particular,

$$\dim(\mathcal{S}_T(H)) = 0 \quad \Longleftrightarrow \quad \text{there is a *unique* matrix } A \in \mathcal{C}(H) \text{ consistent with the trajectory.}$$

**How $\dim(\mathcal{S}_T(H))$ is computed.** Let $r := \dim(\mathcal{C}(H))$ and fix any basis $\{A_1, \ldots, A_r\}$ of $\mathcal{C}(H)$, so that every $A \in \mathcal{C}(H)$ can be written as $A = \sum_{i=1}^r \theta_i A_i$. Then the constraint $Y = AX$ becomes the linear system

$$\mathrm{vec}(Y) = \begin{bmatrix} \mathrm{vec}(A_1 X) & \cdots & \mathrm{vec}(A_r X) \end{bmatrix} \theta,$$

where $\mathrm{vec}(\cdot)$ stacks the columns of a matrix into a single vector. If this system is feasible, then $\dim(\mathcal{S}_T(H))$ equals the dimension of the null space of the above design matrix, i.e., $r$ minus its rank.

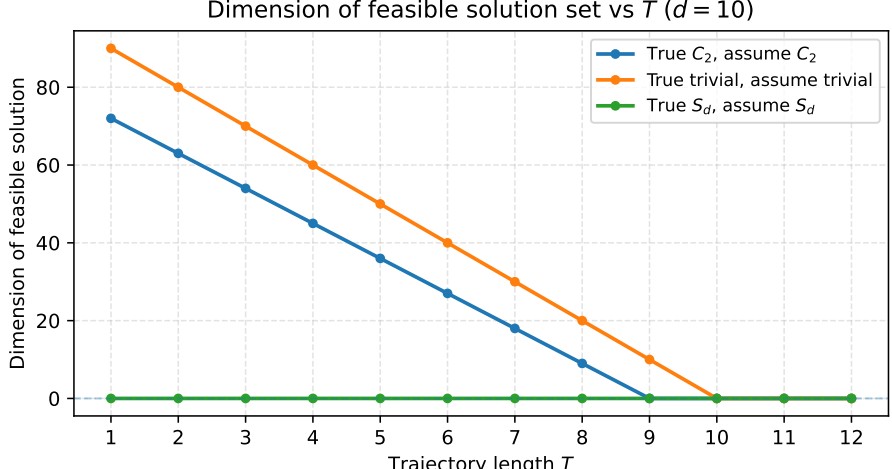

*Figure 1.* Dimension of the feasible solution set as a function of trajectory length.

**Three matched settings.** We set $d = 10$ and sample the initial state as $x_0 \sim \mathcal{N}(0, I_d)$. We repeat trials where the true dynamics matrix $A$ is sampled from: (i) $\mathcal{C}(C_2)$, where $C_2$ is generated by a single coordinate swap, (ii) the unconstrained class $\mathbb{R}^{d \times d}$ (no symmetry), and (iii) $\mathcal{C}(S_d)$, the fully symmetric case. For each trial and each trajectory length $T$, we compute $\dim(\mathcal{S}_T(H))$ under the correctly matched assumption $H$ (namely, $C_2$, trivial, or $S_d$), and report the median over trials.

**Observed behavior.** Figure 1 plots $\dim(\mathcal{S}_T(H))$ versus $T$. As $T$ grows, the number of linear constraints in $Y = AX$ increases, so the feasible-set dimension shrinks and eventually reaches 0, corresponding to unique identification. Stronger symmetry reduces the number of degrees of freedom in $A$ and thus yields uniqueness from shorter trajectories: in our experiment, the fully symmetric case ($\dim(\mathcal{C}(S_d)) = 2$) reaches $\dim(\mathcal{S}_T(S_d)) = 0$ essentially immediately, the single-swap case ($\dim(\mathcal{C}(C_2)) = 82$ for $d = 10$) reaches $\dim(\mathcal{S}_T(C_2)) = 0$ around $T \approx 9$, and the unconstrained case ($\dim(\mathbb{R}^{d \times d}) = 100$) reaches $\dim(\mathcal{S}_T(\{e\})) = 0$ around $T \approx 10$.

# D. Notation

We collect the notation used in the representation-theoretic analysis of the equivariant dynamics.

| Symbol | Meaning |
| --- | --- |
| $G$ | A group acting on the state and feature spaces |
| $\widehat{G}$ | Set of irreducible representations of $G$ |
| $\pi \in \widehat{G}$ | An irreducible representation of $G$ |
| $V_\pi$ | Representation space of the irrep $\pi$ |
| $d_\pi$ | Dimension of $V_\pi$ |
| $\rho$ | Linear representation of $G$ on the state space $\mathbb{R}^d$ |
| $\rho_\Phi$ | Linear representation of $G$ on the feature space $\mathbb{R}^m$ |
| $\Phi$ | Feature map $\Phi : \mathbb{R}^d \to \mathbb{R}^m$ |
| $W$ | Linear dynamics map $W : \mathbb{R}^m \to \mathbb{R}^d$ |
| $x_t$ | State at time $t$ |
| $T$ | Trajectory length |
| $n_\pi$ | Multiplicity of $\pi$ in the state representation $\rho$ |
| $m_\pi$ | Multiplicity of $\pi$ in the feature representation $\rho_\Phi$ |
| $x_{\pi,t}$ | Projection of $x_t$ onto the $\pi$-isotypic component |
| $\Phi_{\pi,t}$ | Projection of $\Phi(x_t)$ onto the $\pi$-isotypic component |
| $W_\pi$ | Restriction of $W$ to the $\pi$-isotypic block |
| $C_\pi$ | Multiplicity-space matrix for the $\pi$-block of $W$ |
| $I_{V_\pi}$ | Identity map on the irrep space $V_\pi$ |

*Table 2.* Summary of notation for equivariant dynamics and isotypic decompositions.

