# OpenReview forum: "Adaptive Symmetry Discovery for Dynamical System Identification"
_ICML.cc/2026/Conference — ICML 2026 regular_

### Official Review · Reviewer_MgcU · 2026-02-18

**Soundness:** 4
**Presentation:** 4
**Significance:** 4
**Originality:** 4
**Overall Recommendation:** 6
**Confidence:** 5

**Summary:**

In many important applications, a dynamically system has underlying symmetries that dictate the possible behaviors of its trajectory. In this work, the authors consider the problem of learning the behavior of a dynamical system in the presence of such symmetry. To this end two major results are proven. The first of which illustrates how this symmetry makes the job of the learner much easier, requiring far fewer samples to make an intellegent hypothesis than in the general case. The second of which is concerned with the question of whether the symmetry group itself can be learned provided enough training data from the system.

**Compliance With Llm Reviewing Policy:**

Affirmed.

**Final Justification:**

My score is unchanged through the discussion period.

This was by far one of the most interesting papers that I was asked to review. The mathematics underlying the work is extremely sophisticated and broad. The writing is well organized and motivated. The results of the work are also very strong.

I saw that a number of reviewers voiced concerns about practical implementation and explicit data collection with the methods discussed in this work. While I can understand these concerns, I am personally less moved by them as I view this paper as a theoretical work first and foremost.

**Key Questions For Authors:**

My only question to the authors is for my own curiosity, and not critical to my evaluation. I often work in contexts where there is either more than one group naturally acting in the background, or otherwise where there are families of systems each of which acted on by a different group. I wonder whether the authors have considered these situations as well? For instance, in the vein of representation theory in machine learning, there is a paper of Levin and Diaz on the any dimensional input problem for equivariant NN's. In that work they show how the presence of higher and higher symmetries can lead to a possibly elegant solution to the any dimensional input problem.

**Limitations:**

Yes

**Strengths And Weaknesses:**

Soundness - The mathematics underlying this work is sound. Proofs or outlines of proofs are included for all major statements, or otherwise given a reference to justify.

Presentation - The presentation of the material is excellent in every regard. Background is presented for completeness, but never becomes overbearing or tedious. Proofs are provided and written often in more intuitive ways, while leaving technical details to appendices. Citations to prior works are generous and give an excellent overview of the field and why what they are doing in this paper is significant.

Significance - The intersection of dynamical systems and learning theory is an extremely hot topic right now, with mathematicians and computer scientists both making significant progress. This work considers what can be gained by noting the presence of symmetry. This is an incredibly natural question in the context of dynamical systems, that has a lot of precedence in other contexts. As one might imagine, actually tackling a problem of this sort will inevitably require the incorporation of group theory and representation theory, which this work does seamlessly.

Originality - As stated in the previous point, the idea of taking a well studied problem and incorporating symmetries is a very common track. That being said, however, the devil is, as usual, in the details. To accomplish the precise estimates performed in this work, the authors needed to juggle a number of highly non-trivial facets of a number of different mathematical disciplines. I personally feel that there are likely very few people who could have done it as well as this paper does.

---

> ### Author Rebuttal · Authors · 2026-03-31
>
> We greatly thank Reviewer MgcU for acknowledging our results, and we are proud and excited to see that they liked our paper. This is a heartwarming signal for our research, and we appreciate the reviewer for giving us the energy to continue research in this direction.
>
>  - Question: connection to representation stability:
>
>
>  Thanks for mentioning this. Yes, we are familiar with representation stability, and it is indeed related to our paper as well. Considering the example we provided in the paper, the permutation symmetries have a "stabilized" parameter space (only four parameters are enough, instead of $d^2$). As $d$ enlarges, this stability leads to the fact that we can solve the problem (here identifying the dynamic) using a number of samples independent of $d$. In the general case of having a growing family of groups, each with a different action, one can take the limit of our quantity $T(G)$ to see if it stabilizes or not. If the sequence of groups stabilizes (under the representation stability assumption), then we have already quantified the gain. It seems we forgot to explain the connection between the representation stability in the paper. Sorry! We will do it in the next version, along with references to the relevant citations in math and ML venues.

---

> > ### Author Rebuttal · Reviewer_MgcU · 2026-03-31
> >
> > The authors have addressed any concerns or questions that I had.

---

> > > ### Author Response · Authors · 2026-04-08
> > >
> > > Thank you for your support of the acceptance!

---

### Official Review · Reviewer_nC22 · 2026-03-07

**Soundness:** 2
**Presentation:** 3
**Significance:** 2
**Originality:** 2
**Overall Recommendation:** 3
**Confidence:** 4

**Summary:**

This paper provides an adaptive discrete symmetry discovery framework for polynomially lifted linear dynamical systems. First, it studies the generic identifiability of equivariant systems and proposes the sample complexity for G-equivariant system which shows that symmetries can largely reduce the trajectory length required to identify a system.
Then, given a single trajectory that is generated and identifiable by a G-equivariant system, this paper proposes an algorithm to automatically discover the hidden symmetry G by performing a feasibility test over a candidate set of groups.

**Compliance With Llm Reviewing Policy:**

Affirmed.

**Final Justification:**

Although the theoretical analysis in this paper is strong, however, the setting considered in this paper appears rather idealized, and analyzing the effect of noise is important for establishing the robustness and practical relevance of the results. Thus, I retain my original score.

**Key Questions For Authors:**

1. For Algorithm 1, is there any specific order of the candidate group set $\mathcal{G}$ predefined in the for loop?

2. Could the author provide a sensitivity analysis about the impact of data noise on Algorithm 1?

**Limitations:**

yes

**Strengths And Weaknesses:**

## Strengths

1. The theory proposed in this paper is elegant. It nicely integrates dynamical system identification with symmetry.

2. The main text is well-written and well-organized.



## Weaknesses

1. In line 900, the row of $C_\pi$ should not be $d$. Since $W_\pi$ is block-diagonal, the row of  $C_\pi$ should be the multiplicity of irreps $\pi$ in the output representation $\rho(g)$.

2. It is unclear why $\rho(g)$ can induce a unique group representation $\rho_\Phi(g)$ under nonlinear polynomially lift. This is an important foundation of the method setting, and a concrete proof is needed.

3. The setting of the dynamic system considered in this paper is limited. Only dynamic systems with equivariant nonlinear feature lift, such as polynomially lifted linear dynamical systems, are suitable in this framework.

4. The framework of this paper is highly idealized, and it faces a lot of challenges in applying this method in a practical setting. In this paper, only synthetic noise-free experiments with known ground truth are carried out.  For a real-world setting, a noisy single trajectory would make it harder for Algorithm 1 to detect the real symmetry. Besides, the framework relies on a lot of predefined hyperparameters, such as candidate group set $\mathcal{G}$, group representation $\rho(g)$, and degree $k$, which also increase the difficulty of handling real-world problems.

---

> ### Author Rebuttal · Authors · 2026-03-31
>
> We thank the reviewer for their valuable comments. Here we provide our response:
>
>
>  - Noisy regime (Question 2):
>
> We can consider the noisy regime, where the actual dynamics is something like $x_{t+1} = f(x_t) + e_i$ for some noise variables $e_i$. Again, under mild conditions, given the results of our paper, we could obtain the estimation rate of order $\frac{p}{T}$, where $T$ is the trajectory length, $p$ is the number of effective parameters. How to prove this? There is a standard literature on estimation in noisy dynamical systems, and we need to mimic this for our case, which is nicely represented as a linearly constrained system, thanks to our algorithm.
>
>   - Continuous groups:
>
>  We considered finite groups as they are combinatorial and look more challenging, yet we solved the problem for them. For continuous groups, instead of using "generators," which are the core idea of the paper, we can use "Lie group generators," which are "directional changes" that change the dynamics equivariantly. This means the problem is no longer combinatorial, and we want to essentially do "subspace recovery". So if we have a Lie group $\tilde{G}$ and we just know that the actual unknown symmetries are a Lie subgroup of $\tilde{G}$, we need to do subspace recovery to find the right symmetries. This is an easier problem, and can be solved in polynomial time, too. We will add this extension to the next version of our paper to make things complete, emphasizing that the finite group version of the problem is more challenging, as it involves combinatorial optimization.
>
>
>  - Brute force search (Question 1):
>
>  We can improve the brute force search to a greedy algorithm: instead of searching over all candidate groups, one can sample elements from a big group $\tilde{G}$ that we seek to find its right subgroup in symmetry discovery, and then do a feasibility check (Algorithm 1). Then, if the program is still feasible, we can sample another point from $\tilde{G}$ and continue. Does this method converge? Yes! And it allows us to recover any subgroup, under a mild condition: the size of the true group of symmetries should be large enough, so that sampling from $\tilde{G}$ hits the target subgroup with good probability.
>
>
> We believe this new viewpoint can address your concerns about brute force, as it replaces it with just greedy. For generic cases (no lower bound on hitting the subgroup via random sampling), we believe this problem is extremely hard computationally. It involved searching over all subsets of a set of size $exp( \Omega(d))$, and this is extremely hard. It is not even clear if this problem is NP hard, as it might be even in a harder complexity class. Under mild conditions, though, we solved the problem, perhaps surprisingly, using a greedy method in polynomial time.
>
>
> Moreover, the order of candidate groups in Algorithm 1 doesn't matter, and we just need a list of them. Algorithm 1 works for any order, and our results remain unchanged.
>
> - Weakness 1:
>
> Thanks a lot for pointing this out. We double-checked the proof, and it is a typo. We will correct the typos in the proof in the next version, and we checked that the statement of the theorem and our algorithm are not affected by that.
>
> - Weakness 2:
>
> The fact that a representation $\rho$ can be lifted uniquely to a representation on the polynomial features can be seen through tensor product: we can consider the tensor power of $\rho$ via the tensor powers of matrices, and then we linearly map it into the space of so-called symmetric tensors, which are in bijection with the polynomial features, thus it gives a unique constructible representation. We will add the explanation to the paper.
>
> - Weakness 3:
>
> Our setting, while being limited, is one of the most general cases where learning dynamics are possible, provably and efficiently. We believe our setting is sufficiently generic for a theory-oriented paper, and we will explain this future direction in the next version of the paper.
>
>
>
> - Weakness 4:
>
>
> Answered in the above cases.
>
>
>
>  Lastly, we wanted to emphasize that, given the time constraints of the rebuttal week and the character limit, we kept this response concise. We are happy to engage in further discussion and provide more details. Finally, we see that other than your overall recommendation, other scores are low. We would be happy to address your concerns for a potential improvement in such scores.

---

> > ### Author Rebuttal · Reviewer_nC22 · 2026-04-03
> >
> > Thanks for the rebuttal. However, I think two issues are still not resolved.
> >
> > First, for the noisy dynamical system setting, the rebuttal only says that a similar analysis should be possible by following standard literature. But it does not give a precise noise model, assumptions, modified theorem, or experiment results. So this point is still not really addressed.
> >
> > Second, I do not agree that the order of candidate groups is unimportant. If Algorithm 1 stops once it finds the first group that passes the test, then it may stop at a proper subgroup of the true group. In that case, the algorithm would return the subgroup instead of the full true group. So the output can depend on the search order. To avoid this issue, the authors would need at least to specify a search order under which the algorithm is guaranteed to recover the true group.

---

> > > ### Author Response · Authors · 2026-04-08
> > >
> > > Thanks for the follow-up questions. Due to character limits in the rebuttal, we could not include all details previously. We clarify below.
> > >
> > > > Noisy regime (and references):
> > >
> > > For the noisy setting, we refer to Theorem 1 in [1] (already cited in the paper, line 60, col. 2). They consider $x_{t+1}=Ax_t+e_t$ with $e_t$ i.i.d. isotropic Gaussian noise (extendable to subgaussian noise; see Assumption 1 in [1]) and show that the least-squares estimator satisfies $\|A-\hat A\|_2^2 \le C_d/T$, where $T$ is the trajectory length and $C_d$ depends on the dimension. Comparing with Eq. (2) in [1], our estimator is essentially the same. After symmetry discovery, one can project onto the equivariant class and invoke [1] to obtain the corresponding rate.
> > >
> > > Extending this reduction rigorously to the noisy setting requires Cayley-graph steps, which we omit for space. Our method yields $\|A-\hat A\|_2^2 \le C_d C_G/T$, where $C_G = T(G)/T(\text{no symmetry})$ (as defined in the paper). In the noisy regime, the feasibility test is replaced by a least-squares objective $\|WX-Y\|^2$ under equivariance constraints. If enforcing symmetry reduces the loss, the symmetry is retained; otherwise, it is rejected.
> > >
> > > [1] Near-optimal finite-time identification of arbitrary linear dynamical systems, ICML 2019
> > >
> > > > Order of candidate groups
> > >
> > > Yes, that is correct. In our analysis, we implicitly assume that only one candidate group survives the feasibility test in Algorithm 1. To recover all such groups, one would need to loop over all candidates and select the maximal subgroup. However, this can be done more systematically via a greedy approach.
> > >
> > > > Greedy method (new):
> > >
> > > To avoid enumerating all candidate groups, we propose a greedy procedure. Let $G'$ be a known ambient group and suppose the true symmetry group is an unknown subgroup of $G'$. Initialize $S=\emptyset$. At each step, sample $g\in G'$ and test feasibility in Algorithm 1 (Line 6) for $S\cup\{g\}$. If feasible, add $g$ to $S$; otherwise discard it.
> > >
> > > If sampling hits the true subgroup with probability at least $p$, then standard results on Cayley expanders imply that $O((\log |G'|)/p)$ iterations suffice to recover the symmetry, providing a scalable alternative to brute-force enumeration. Importantly, this procedure progressively enlarges the discovered subgroup: as $S$ grows, so does the discovered symmetry, and after $O((\log |G'|)/p)$ steps the full symmetry group is recovered.
> > >
> > > > Conclusion
> > >
> > > Lastly, we hope you find our responses sufficient to support a revision and an acceptance recommendation. Within the character limits and time constraints, we hope we have addressed all concerns and will incorporate these clarifications in the revised version of the paper.

---

### Official Review · Reviewer_6cDA · 2026-03-12

**Soundness:** 3
**Presentation:** 4
**Significance:** 2
**Originality:** 3
**Overall Recommendation:** 5
**Confidence:** 2

**Summary:**

The paper discusses theoretical results on identifying dynamical systems that are also symmetric (the dynamical rules are invariant under the action of a symmetry group $G$). For technical reasons, the paper restricts itself to discrete-time-step systems, with polynomial right hand sides. The state can be a continuous real vector.

The paper then looks at two problems: (i) How much easier or more difficult in terms of sample complexity (using a trajectory of $T$ steps as "training data") becomes the problem of determining the right hand side (identifying the dynamical rules) when the law is symmetric under a finite symmetry group? (ii) How costly/difficult is it to in addition select on group $G$ out of a finite set of admissible candidate groups $\mathcal{G}$?

The first question yields the answer that symmetry actually reduces the sample complexity as it puts an additional subspace constraint on the (linear) dynamical map in polynomial feature space. The second question is answered by an algorithm that checks the groups one-by-one (linear time) and sampling only a logarithmic sized set of candidate generators, and the appendix shows that this is sufficient to find the correct group with high probability.

The paper does not provide concrete applications yet, but the appendix discusses a short practical experiment and implementation on an abstract toy case (showing that the ideas work in "practice" in an algorithmic sense).

**Compliance With Llm Reviewing Policy:**

Affirmed.

**Final Justification:**

The rebuttal was helpful with clarifying several technical question, in particular about the group complexity arguments (where my reasoning laid out above that doubts the papers argument was actually wrong). It thus strengthened my already very positive perception (I already gave a high score being optimistic that issues could be clarified; so I do not see a need to raise it at this point).

In terms of the overall assessment by the other reviewers, I do not feel knowledgeable enough in this sub field to synthesize the diverging views. I would keep my high score as personal opinion but would also point out that my confidence in my rating and understanding of the broader subject matter is limited.

**Key Questions For Authors:**

A central argument for the second theorem (4.4) and the corresponding randomized algorithm is that the groups are generated by a small set of generators. It is stated explicitly in ll.410ff: "a random subset of a finite group G of size O(log |G| + log 1/δ ) forms a generating set with probability at least 1 − δ".
I do not see why this should hold in this generality. Assume a very simple group which has the action of flipping a system between a set of binary states, each individually (so it is the direct product of small groups, such as $(F_2,+)^n$. Such a group would have a set of generators linear in the number of group elements (n= |G|/2 in the example), as each "flip" is independent of each other. Why should there thus always be a logarithmic generating subset? I might be misunderstanding something fundamentally here - I am not a mathematician - or maybe some additional constraint (arising from coupling to the dynamical system) is used (later in the appendix, which I did not fully understand).

A few more very minor things:
- In ll. 168 (right), are $k$ and $T$ identical/the same variable?
- In Algorithm 1, Step 5: $N$ should be rounded here?
- The paper does not explain why trajectories are used for identification instead of random time-step samples, which would seem easier to understand. The appendix addresses the problem of generality of trajectories; maybe the main paper could mention the key aspect briefly (and prominently).

**Limitations:**

One could discuss the structural limitations (linear in number of groups, discrete symmetry) and possible implications, and maybe work-arounds a bit. I do not see unaddressed societal issues.

**Strengths And Weaknesses:**

First, a short disclaimer: I know group theory and dynamical systems only from an "engineering" perspective; some technical details are beyond my knowledge, so I cannot vouch for the correctness of the details of the proofs.

Strength:
- Despite being a formal and theoretical topic, I found the paper easy and enjoyable to read. Except from some technical details of the proofs, using spectral graph theory on Caley graphs, even most of the appendix seemed accessible and very well written to me. The structural split in a high-level view in the paper and details reiterated in the appendix felt well designed and appropriate for the topic.
- The paper does provide optimal solutions to the claimed problems.
- The basic ideas and arguments are mostly intuitively accessible and plausible.
- I am not aware of prior work on this specific problem, but I am really not an expert in this branch of theory.

Weaknesses:
- (minor) The paper at this point cannot yet make a strong case for utility in either applications of its algorithms or understanding/modeling of conceptual questions of direct practical relevance; it is pure theory (nothing wrong with that, but it is a limitation in terms of direct impact).
- (minor) Some important aspects are only presented in the appendix; the paper should reference the appendix explicitly (easy to fix, but confusing on first read). I would suggest summarizing the experimental validation briefly, too, as this makes the paper more well-rounded.
- (major) Some of the assumptions and limitations are quite strong. Being linear in $|\mathcal{G}|$ can be a serious obstacle in practical scenarios where we might be searching for symmetry from a very large class of options. The discrete nature of $G$ and $\mathcal{G}$, similarly, excludes many important practical applications (such as simple Newtonian Physics, where we of course know the invariances). One could probably lift the discrete model to build approximate search strategies, but this is not addressed so far.
- (medium/partially fixable) It would be nice to have a bit more experimental validation; one could at least move some results from the appendix into the main paper.
- (maybe mayor, most likely none) There were some aspects that I did not understand, see questions below.

Overall, I like the paper because it makes a very clear and systematic case for its findings. As far as I am able to understand and verify the theory, the only weakness seems to be that the problem setting is still somewhat detached from practical applications, with some strong formal restrictions. I should emphasize that am not able to assess novelty in the context of the broader literature, and while the arguments appeared mostly plausible to me, I am not in a position to formally verify the proofs.

---

> ### Author Rebuttal · Authors · 2026-03-31
>
> We extremely thank Reviewer 6cDA for the acknowledgement of our results, and we are proud and excited to see that they liked our paper.
>
>  - Weakness 1:
>
> We agree with the reviewer that our results are theory-oriented, as our method does not have linear runtime (like gradient methods) that are scalable in practice.
> The next step is to use the expander ideas in more practical cases to see if we can achieve optimal sample complexity with linear runtime. This is an interesting future direction, demanding  to integrate expander methods into practical methods. We will add the explanation to the paper.
>
>  - Weakness 2: Stuff presented in the appendix:
>
>  Thanks for mentioning this. We will bring them into the main body as we will have more space later for the next version.
>
>  - Weakness 3: Strong assumptions
>
> NEW: Indeed, we can relax the strong assumptions! Here is our explanation:
>
>   - Extending to Continuous groups:
>
>  We considered finite groups as they are combinatorial and look more challenging, yet we solved the problem for them. For continuous groups, instead of using "generators," which are the core idea of the paper, we can use "Lie group generators," which are "directional changes" that change the dynamics equivariantly. This means the problem is no longer combinatorial, and we want to essentially do "subspace recovery". So if we have a Lie group $\tilde{G}$ and we just know that the actual unknown symmetries are a Lie subgroup of $\tilde{G}$, we need to do subspace recovery to find the right symmetries. This is an easier problem, and can be solved in polynomial time, too. We will add this extension to the next version of our paper to make things complete, emphasizing that the finite group version of the problem is more challenging, as it involves combinatorial optimization.
>
>  - Avoiding Brute force search:
>
>  We can improve the brute force search to a greedy algorithm: instead of searching over all candidate groups, one can sample elements from a big group $\tilde{G}$ that we seek to find its right subgroup in symmetry discovery, and then do a feasibility check (Algorithm 1). Then, if the program is still feasible, we can sample another point from $\tilde{G}$ and continue. Does this method converge? Yes! And it allows us to recover any subgroup, under a mild condition: the size of the true group of symmetries should be large enough, so that sampling from $\tilde{G}$ hits the target subgroup with good probability.
>
> We believe this new viewpoint can address your concerns about brute force, as it replaces it with just greedy. For generic cases (no lower bound on hitting the subgroup via random sampling), we believe this problem is extremely hard computationally. It involved searching over all subsets of a set of size $exp( \Omega(d))$, and this is extremely hard. It is not even clear if this problem is NP hard, as it might be even in a harder complexity class. Under mild conditions, though, we solved the problem, perhaps surprisingly, using a greedy method in polynomial time.
>
> - Weakness 4:
>
>  As we detailed to another reviewer, we can extend our results to the noisy regime (not mentioned by you, that's why we didn't detail it here). This means we can also extend our experiments to cases where we have noise, with the same algorithm (replacing identification with least squares). Our proposed method is fairly simple to implement in mid-scale as it only involves linear algebraic methods. It is a bit tight to complete them here in the rebuttal week and report them back to you, but we will certainly include them in the next version of the paper to complete the story.
>
> - Question: Binary sign-flips in high dimension
>
> Thank you for mentioning this, as it is an excellent question to add to the paper for its next version. Regarding your concern, in such a case, the group $\{ \pm 1\}^d$ has $|G| = 2^d$ elements, while if we consider sign flips for each coordinate, we have a set of only $d$ elements generating the group. As a result, in that case, we exactly need $log |G| = d$ size sets to achieve a generative set, consistent with our random sampling method.
>
> - Question $T$ vs. $k$
>
> Thanks for mentioning the ambiguity. They are different variables; $T$ is the length of the trajectory, and $k$ is the degree of the polynomial we use for lifting to the feature space. We realized that we didn't define $k$ in the definition the reviewer referred to, and we will clarify this in the next version.
>
> - Question: rounding $N$
>
> Yes, we need to round it to the smallest positive integer greater than or equal to it. We will correct this in the next version.
>
> - Question: generality of trajectories
>
> We have a bit of trouble understanding your comment, but if it is concerning the genericity of trajectories that one can achieve with random samples, we completely agree with you, and we can bring more explanations from the appendix to the main body as we will have more space in the next version of the paper.

---

> > ### Author Rebuttal · Reviewer_6cDA · 2026-04-03
> >
> > The rebuttal resolves all of my concerns (including finding a bug in my "counter example" argument). To answer to the last point: Yes, the issue about the trajectories was just to mention the discussion from the appendix.

---

> > > ### Author Response · Authors · 2026-04-08
> > >
> > > Thank you for your support of the acceptance. We will incorporate all the discussed changes into the next version of the paper.

---

### Official Review · Reviewer_ekxu · 2026-03-13

**Soundness:** 2
**Presentation:** 1
**Significance:** 2
**Originality:** 2
**Overall Recommendation:** 2
**Confidence:** 3

**Summary:**

This paper proposes an algorithm that reduces the required number of trajectories when polynomially fitting the dynamics, using brute-force symmetry search. The symmetry is parameterized by a subgroup of a discrete group, acting on the state space. The paper shows how the required degree of freedom is reduced by enforcing the symmetry, in representation-theoretical background. The algorithm is not validated by experiments.

**Compliance With Llm Reviewing Policy:**

Affirmed.

**Final Justification:**

Although I agree the setting discussed in this paper can be widened by covering noisiness, continuous symmetries and different feature lifting, many of the arguments are still hypothetical. I maintain my score.

**Key Questions For Authors:**

See weaknesses.

**Limitations:**

yes

**Strengths And Weaknesses:**

- strengths
  - This paper explains how the space of polynomials can be drastically reduced when symmetries are enforced, in representation-theoretical perspective. The paper built an actual upper bound for the required number of trajectories, and showed an example that the number is actually reduced by enforcing the symmetries (e.g., Example 4.2).
- weaknesses
  - The method is not validated with any experiments. This paper does not address how experimentally feasible the proposed algorithm is.
This method assumes the dynamics function $f$ is a polynomial and $x$. In most cases, this is an approximation, unless one finds a nice state space such that the dynamics are easily represented (e.g. a Koopman operator). The proposed algorithm only works when the (polynomial) dynamics are \textit{perfectly identifiable by polynomials}. The method will be much applicable to broader domains if it concerns the case $f(x) \approx W \Phi(x)$.
  - This paper assumes $\mathcal{G}$ is a discrete group. Many groups that act on the physical space are continuous groups, e.g., GL(n).
  - The idea that “use symmetry for reducing the degree of freedom for system identification” is not novel. In fact it’s what we do everyday – although it’s plausible that this paper managed to formalize the representational theoretical background of that idea.
    - Let $f(x)$ be a 1-dim scalar function. Then $f(x) = a_0 + a_1x + a_2 x^2+\cdots$. But if we know $f(x) is even, then $f(x) = a_0+ a_2 x^2+\cdots$.
    - Let’s bring the Example 4.2 (Quadratic systems with permutation symmetry and let $x_1,...,x_4$ be the four variables. Since we assume quadratic dynamics, the function $f$ will be a linear combination of $1, x_1,x_2…,x_1^2,...,x_1x_2,...$, and permutation symmetry enforces $f$ a linear combination of $1, \sum_i x_i, \sum x_i^2, \sum x_ix_j$.
  - If we’re going to do full polynomial expansion of a multidimensional system, then the number of unknown parameters grows exponentially, since $\binom{d+k}{d} = \Theta(d^k)$. Although the method claims that it can reduce the number of observations using symmetries, I’m unsure how scalable it will be when the dynamics are multidimensional and $k$ is large.
  - I’m not sure where the term 2.67(...) came from, in Algorithm 1. I think those terms appearing in the main pseudocode should be explained in the main paper, not in the appendix.

---

> ### Author Rebuttal · Authors · 2026-03-31
>
> We thank the reviewer for their valuable comments. Here we provide our response:
>
>
>  - Polynomially lifted features and noisy regime:
>
>  We consider polynomials just as an instance of lifting to the feature space. One can consider, e.g., Fourier features, or general finite-dimensional kernels, and the proof follows along the same lines (under the mild condition that the features are well excited so that we can identify them). Nothing in the results changes, and so we will modify the next version of our paper to consider this case.
>
>  Moreover, one can also consider the noisy regime, where the actual dynamics is something like $x_{t+1} = f(x_t) + e_i$ for some noise variables $e_i$. Again, under mild conditions, given the results of our paper, we could obtain the estimation rate of order $\frac{p}{T}$, where $T$ is the trajectory length, $p$ is the number of effective parameters. How to prove this? There is a standard literature on estimation in noisy dynamical systems, and we need to mimic this for our case, which is nicely represented as a linearly constrained system, thanks to our algorithm.
>
>
>  - Experiments:
>
>  We already have a small experiment in the appendix of the paper. Our proposed algorithm does not involve challenging procedures as it just needs sampling from a group, we will certainly do add more experiments to the next version of our paper. The current setting we considered in the paper experiment section shows the algorithm works, and we will extend it in the next version.
>
>
>  - Continuous groups:
>
>  We considered finite groups as they are combinatorial and look more challenging, yet we solved the problem for them. For continuous groups, instead of using "generators," which are the core idea of the paper, we can use "Lie group generators," which are "directional changes" that change the dynamics equivariantly. This means the problem is no longer combinatorial, and we want to essentially do "subspace recovery". So if we have a Lie group $\tilde{G}$ and we just know that the actual unknown symmetries are a Lie subgroup of $\tilde{G}$, we need to do subspace recovery to find the right symmetries. This is an easier problem, and can be solved in polynomial time, too. We will add this extension to the next version of our paper to make things complete, emphasizing that the finite group version of the problem is more challenging, as it involves combinatorial optimization.
>
>
> - The idea of using symmetry is not novel:
>
> Our contribution is to (1) quantify the gain, with a closed-form representation theoretic formula, (2) propose an algorithm for symmetry discovery to achieve the optimal sample complexity. The idea that symmetry reduces the parameter space is intuitive, and it is neither clear to what extent it can help, nor if one can use it in symmetry discovery. We solved both.
>
> - Exponentially many parameters:
>
> If both $d$ and $k$ are large, there is not much we can do ever. This becomes the problem of learning an arbitrarily high-dimensional feature space, and as far as we know, the curse of dimensionality is unavoidable. Nevertheless, if $d$ is large but $k$ is fixed, the dimension of equivariant dynamics parameters can stabilize as a function of $k$.  Example: we already gave an example of stability in our paper, as the dimension doesn't grow with $d$ for permutation invariant dynamics. This is not restricted to the permutation group and polynomial features, and it is called "representation stability" in the literature. As a result, we can find solutions to a problem that naively needs arbitrarily large trajectories, with much smaller trajectories after we take care of symmetry discovery. That being said, it is all about trajectory length, as one may want to apply a cheap method, such as gradient methods, to obtain not only polynomial but indeed linear runtime. This is an interesting direction, and it falls outside of the current scope of this paper (our goal is to achieve the smallest trajectory length + polynomial runtime). We mention this as a future direction in the next version of the paper.
>
>  - The term in Algorithm 1
>
>  We will bring more detail to the main body of the paper in the next version. The constant 2.67 is indeed 8/3, coming from a standard concentration inequality exponent constant in the literature.
>
>   - Brute force search:
>
> We addressed this issue as well, using a greedy algorithm (the full response to this is provided in another reviewer's rebuttal, please see other rebuttals for space constraints)
>
>
>  Lastly, we wanted to emphasize that since we are limited to the rebuttal week time constraints, as well as the character number constraints, we made this response concise. We are happy to engage in further discussion and provide more details. Finally, we see that other than your overall recommendation, other scores, such as presentation, are low. We would be happy to address your concerns for a potential improvement in such scores.

---

> > ### Author Rebuttal · Reviewer_ekxu · 2026-04-04
> >
> > Thank you for your response.
> >
> > Polynomially lifted features and noisy regime
> > - I believe such a dynamics identification method should consider a noisy regime, because in many realistic cases, the feature lifting will not guarantee exact identifiability.
> > - The author mentioned the polynomials as an instance of lifting – if the lifted features are not polynomials, does the representation theory extend easily like the decomposition in Theorem 4.1? The group action may not be linear in the lifted features, if they are not polynomials.
> > - The author mentioned “standard literature”. Can you give me references?
> >
> > Continuous groups
> > - The author only provided a high-level explanation instead of mathematically written arguments.
> >
> > Empirical usefulness of the proposed algorithm
> > - The theorem 4.4 states the runtime is polynomial in $|\mathcal{G}|$. If G is a permutation group, it will be $n!$. Is it useful, i.e. does the reduction of the trajectory length compensate for this?

---

> > > ### Author Response · Authors · 2026-04-08
> > >
> > > Thanks for the follow-up questions. Due to character limits in the rebuttal, we could not include all details previously. Here we clarify.
> > >
> > > > General features
> > >
> > > Let $\rho$ denote the group action on $x\in\mathbb{R}^d$. Let $\phi_1,\dots,\phi_m$ be linearly independent features and define $\mathcal H=\mathrm{span}(\phi_1,\dots,\phi_m)$. Assume $\mathcal H$ is closed under the group action, i.e., $f\in\mathcal H \Rightarrow f(gx)\in\mathcal H$ for all $g\in G$.
> > >
> > > Then $\rho$ induces a unique linear representation on $\mathcal H$: for each $g\in G$, there exists $A_g\in\mathbb{R}^{m\times m}$ such that $\Phi(gx)=\rho_\Phi(g)\Phi(x)$ with $\Phi(x)=(\phi_1(x),\dots,\phi_m(x))$. Thus Eq. (3) holds and Theorem 4.1 applies. The key point is closure: each $\phi_i(gx)$ is a linear combination of the $\phi_j$.
> > >
> > > Example. Consider Fourier features $\phi_w(x)=e^{2\pi i\langle w,x\rangle}$ for $w\in\mathbb{Z}^d$ with
> > > $\|w\|_2\le C$
> > >
> > > (band-limited functions). For any permutation matrix $A$, $\phi_w(Ax)=e^{2\pi i\langle w,Ax\rangle}=e^{2\pi i\langle A^\top w,x\rangle}=\phi_{A^\top w}(x)$. Since $A^\top=A^{-1}$ is again a permutation matrix, $\|A^\top w\|_2=\|w\|_2\le C$. Hence this feature space is closed under permutations (and all subgroups).
> > >
> > > More generally, Laplacian eigenspaces on manifolds (Fourier modes) are invariant under isometries and satisfy the same closure condition.
> > >
> > > > Empirical usefulness
> > >
> > > Note that $|\mathcal G|$ denotes the number of candidate groups (line 200, col. 2) and is not the size of a group (not $n!$). The runtime is linear in $|\mathcal G|$ (line 1033), due to the explicit loop over candidate groups. Can this be improved? Yes, please see below for a greedy approach that avoids enumerating all elements of $\mathcal G$.
> > >
> > > Note that the reduction in trajectory length can yield arbitrarily large sample complexity gains (Example 4.2). The gain grows with the dimension.
> > >
> > > > Continuous groups
> > >
> > > Let $G$ be a Lie subgroup of $O(d)$ acting linearly on $x\in\mathbb{R}^d$ (lifted features behave similarly; omitted due to space). In Algorithm 1, we check feasibility of $WX=Y$ under constraints $gW=Wg$ for all $g\in G$. For continuous groups, it suffices to enforce commutation with generators: if $A_1,\dots,A_L$ span the Lie algebra of $G$, then imposing $A_iW=WA_i$ for all $i$ is equivalent. Since $L\le d^2$, one can adapt the finite-group argument (with $L$ replacing $\log|G|$) to recover the correct symmetry in Algorithm 1.
> > >
> > > > Noisy regime (and references)
> > >
> > > For the noisy setting, we refer to Theorem 1 in [1] (already cited in the paper, line 60, col. 2). They consider $x_{t+1}=Ax_t+e_t$ with $e_t$ i.i.d. isotropic Gaussian noise (extendable to subgaussian noise; see Assumption 1 in [1]) and show that the least-squares estimator satisfies $\|A-\hat A\|_2^2 \le C_d/T$, where $T$ is the trajectory length and $C_d$ depends on the dimension. Comparing with Eq. (2) in [1], our estimator is essentially the same. After symmetry discovery, one can project onto the equivariant class and invoke [1] to obtain the corresponding rate. A rigorous reduction in the noisy setting involves technical steps on Cayley graphs, which we omit here due to space constraints.
> > >
> > > [1] Near-optimal finite-time identification of arbitrary linear dynamical systems, ICML 2019
> > >
> > > > Greedy method (new)
> > >
> > > To avoid looping over all candidate groups, we propose a greedy procedure. Let $G'$ be a known ambient group and suppose the true symmetry group is an unknown subgroup of $G'$. Initialize $S=\emptyset$. At each step, sample $g\in G'$ and test feasibility in Algorithm 1 (Line 6) for $S\cup\{g\}$. If feasible, add $g$ to $S$; otherwise, discard it.
> > >
> > > If sampling hits the true group with probability at least $p$, then standard results on Cayley expanders imply that $O((\log |G'|)/p)$ iterations suffice to recover the symmetry, yielding a scalable alternative to brute-force search. The only requirement is access to an ambient group $G'$ for sampling.
> > >
> > > > Conclusion
> > >
> > > Lastly, we hope you find our responses sufficient to support a revision and an acceptance recommendation. Within the character limits, we have addressed all concerns and will incorporate these clarifications in the revised version.

---

### Decision · Program_Chairs · 2026-04-30

**Decision:**

Accept (regular)

**Comment:**

The paper proposes a method for identifying the parameters of a dynamical system by leveraging the symmetry of the system both when the symmetry is known a priori and by discovering the symmetry directly from data. Reviewers were split in their scores, but generally agreed about the strengths and weaknesses of the paper.  Namely, the theory was generally agreed to be sound, well-explained, and sometimes elegant.  However, the paper includes very limited practical empirical validation and the original draft has strong constraints and assumptions which make it unclear if the approach would be applicable in practical settings.  Namely, the dynamics are assumed to be polynomial and the symmetries discrete (although the authors argue their framework also applies to continuous groups in the rebuttal).  The concerns are valid and the critiques of the reviewers are valuable.  At the same time, the theory is sound and will likely be of interest.